# Dynamic molecular changes during the first week of human life follow a robust developmental trajectory

Amy H. Lee [1], Casey P. Shannon [2], Nelly Amenyogbe[3,25], Tue B. Bennike [4,5,6], Joann Diray-Arce[5,6], Olubukola T. Idoko[7,8], Erin E. Gill[1], Rym Ben-Othman[9], William S. Pomat[10], Simon D. van Haren[5,6], Kim-Anh Lê Cao [11], Momoudou Cox[7], Alansana Darboe [7], Reza Falsafi[1], Davide Ferrari[11], Daniel J. Harbeson[3], Daniel He [2], Cai Bing[9], Samuel J. Hinshaw[1,12], Jorjoh Ndure[7], Jainaba Njie-Jobe[7], Matthew A. Pettengill[5], Peter C. Richmond[13,14], Rebecca Ford[10], Gerard Saleu[10], Geraldine Masiria[10], John Paul Matlam[10], Wendy Kirarock[10], Elishia Roberts[7], Mehrnoush Malek[15], Guzmán Sanchez-Schmitz[5,6], Amrit Singh [2,16], Asimenia Angelidou[5,6,17], Kinga K. Smolen[5,6], The EPIC Consortium[#], Ryan R. Brinkman[15,18], Al Ozonoff[5,6,19], Robert E.W. Hancock [1], Anita H.J. van den Biggelaar[14], Hanno Steen [4,5,7], Scott J. Tebbutt [2,20,21], Beate Kampmann[7,22], Ofer Levy [5,6,23] & Tobias R. Kollmann [3,9,25]

Systems biology can unravel complex biology but has not been extensively applied to human newborns, a group highly vulnerable to a wide range of diseases. We optimized methods to extract transcriptomic, proteomic, metabolomic, cytokine/chemokine, and single cell immune phenotyping data from <1 ml of blood, a volume readily obtained from newborns. Indexing to baseline and applying innovative integrative computational methods reveals dramatic changes along a remarkably stable developmental trajectory over the first week of life. This is most evident in changes of interferon and complement pathways, as well as neutrophil-associated signaling. Validated across two independent cohorts of newborns from West Africa and Australasia, a robust and common trajectory emerges, suggesting a purposeful rather than random developmental path. Systems biology and innovative data integration can provide fresh insights into the molecular ontogeny of the first week of life, a dynamic developmental phase that is key for health and disease.

---

[#]A full list of affiliations appears at the end of the paper. These authors contributed equally: Amy H. Lee, Casey P. Shannon, Nelly Amenyogbe, Tue B. Bennike, Joann Diray-Arce, Olubukola Idoko. These authors jointly supervised this work: Anita H. J. van den Biggelaar, Hanno Steen, Scott J. Tebbutt, Beate Kampmann, Ofer Levy, Tobias R. Kollmann.

The first week of life is characterized by heightened susceptibility to infections and is increasingly recognized as a major determinant of overall health for the entire human lifespan[1,2]. Knowledge of the molecular drivers involved in these processes in newborns (defined as those <28 days of life) is fragmentary. Systems biology approaches, employing high-dimensional molecular and cellular measurements (henceforth referred to as OMICs), along with unbiased analytic approaches, have increased our understanding of basal and altered molecular states in adults[3] and recently in newborns and infants after the first week of life[4,5], but such approaches have not been applied systematically to characterize molecular ontogeny over the most critical period, i.e. the first week of life[1]. This is likely due to the analytical challenges posed by the limited amount of biosample that can be obtained[6–8] and the many rapid physiological changes around birth[1]. The resulting variance in biological measurements has been thought to necessitate a large participant sample size, which would increase complexity and cost[9].

To overcome these hurdles, we developed a robust experimental and analytical approach feasible with <1 ml of newborn blood. Our data represent the most comprehensive systems biology study yet performed during the first week of human life. Despite substantial between-subject variation, normalizing (indexing) all samples from a given newborn enabled identification of consistent and robust changes over the first week of life across the entire cohort. Furthermore, data integration using independent strategies not only validated signatures across methodologically- and biologically-distinct datasets, but also provided novel findings. The major observations derived from a cohort from West Africa (The Gambia) were validated for an Australasian (Papua New Guinea) cohort. Our results highlight that, contrary to the relatively steady-state biology observed in healthy adults[7,10], the first week of human life is highly dynamic. Nevertheless, despite the substantial variability between participants and these dramatic changes, ontogeny followed a robust trajectory common to newborns from very different areas of the world.

## Results

**Blood processing**. One of the objectives of this project was to develop a robust standard operating protocol (SOP) to enable extraction and analysis of data using systems biology (big data) approaches from small blood sample volumes that can readily be obtained for research purposes from newborns within the first week of life (Fig. 1, see also Protocol). Our experimental SOP utilized important sample-sparing modifications whereby we obtained samples for immune phenotyping, transcriptomic, proteomic, and metabolomic analysis from <1 ml of blood (see also Supplementary Methods)[11]. We profiled the peripheral blood of each participant twice over their first week of life, i.e. at Day of Life (DOL) 0 (baseline) and additionally at either DOL1, 3, or 7, and sought to identify variables that differed between the baseline and later time points across all participants. This required indexing either by employing paired statistical tests for univariate analyses or calculating fold changes relative to the DOL0 sample for multivariate analyses, as described in online Methods. The number of samples used in each OMIC platform as well as the analysis stage are shown in Supplementary Table 1 and Supplementary Figure 9.

**Immune phenotyping across first week of life**. Determining the cellular composition of biological samples is important in systems biology, as relative and absolute cell numbers predict endpoints of interest with high accuracy, e.g. vaccine responses[12], and enable deconvolution of OMICs data[9,13]. Analysis of our predefined targeted cell populations revealed substantial between-subject variability in peripheral blood samples obtained over the first week of life (Fig. 2, Supplementary Figure 2, Supplementary Note 1). However, consistent within-subject changes over the first week of life amongst the entire cohort of 30 Gambian newborns emerged when samples were indexed as displayed by principal component analysis (PCA). Univariate analysis identified the following discriminating cellular features over the first week of life: basophils, plasmacytoid dendritic cells (DC), natural killer cells, and neutrophils decreased; in contrast, myeloid DCs increased after DOL0, while many other cell types remained stable. We also detected dramatic but consistent changes in soluble immune markers including plasma cytokines and chemokines over the first week of life (Fig. 2). Based on the relevant univariate analysis, we found that plasma concentrations of C−X−C motif chemokine 10 (*CXCL10*), interleukin (*IL*)-17A, macrophage-derived chemokine (*MDC*), and interferon (*IFN*)γ increased, while *IL-10*, Chemokine C−C motif ligand (*CCL*) 5, granulocyte colony stimulating factor 2 (*GCSF*), and *IL-6* decreased with age over the first week of life; many other soluble immune markers remained unchanged.

**Transcriptomic analysis across first week of life**. Modern systems biology studies employ gene expression analysis by RNA-Seq[7,14]. We found that ≥500 μl of adult blood was required to consistently obtain sufficient high-quality RNA, but <100 μl of newborn blood sufficed (Supplementary Figure 3). This likely reflected the relatively high content of white blood cells (WBC) and nucleated red blood cells in newborn blood, which contain abundant globin mRNA[15]. The higher yield and quality of total RNA extracted from newborn vs. adult whole blood was confirmed across different RNA extraction platforms (RNALater and PAXGene; both of which yielded similar results) with the former chosen for subsequent studies (Supplementary Methods).

As with the immune phenotyping data, there was substantial between-subject variability in our RNA-Seq data from peripheral blood samples obtained over the first week of life. This was resolved by indexing each participant to their own DOL0 sample, revealing dramatic yet consistent developmental signals that related to age (i.e. ontogeny) to emerge across the entire cohort. We were concerned that the above-mentioned changes in cell composition could provide a basis for the observed differentially expressed (DE) genes, but the inclusion of cell composition data in our model did not affect our results (see Methods (Online) and Supplementary Figure 3 for details) demonstrating that the whole blood transcriptomic signals driving the developmental trajectory observed were not merely a consequence of changes in the underlying cellular composition across DOL.

In comparing DOL1 vs. DOL0, there were few (12) identified DE genes; however, dramatic developmental changes emerged when comparing later days of life to DOL0 (Fig. 3a). Specifically, for DOL3 vs. DOL0 we detected 1125 DE genes, while on DOL7 vs. DOL0, there were 1864 DE genes. All DE genes, pathway enrichment and statistics are listed in Supplementary Data 5 and Supplementary Methods. In particular, genes with decreased expression across the first week of life are involved in cellular responses to stress, detoxification of reactive oxygen, as well as heme biosynthesis and iron uptake. Conversely genes involved in interferon signaling, Toll-like-receptor (*TLR*), negative regulation of Retinoic Acid Inducible Gene I (*RIG-I*) and complement activation were upregulated over the first week of life.

**Proteomic analysis across first week of life**. A total of 684 different proteins were identified across peripheral blood plasma samples (false discovery rate (FDR) < 1%) (Supplementary

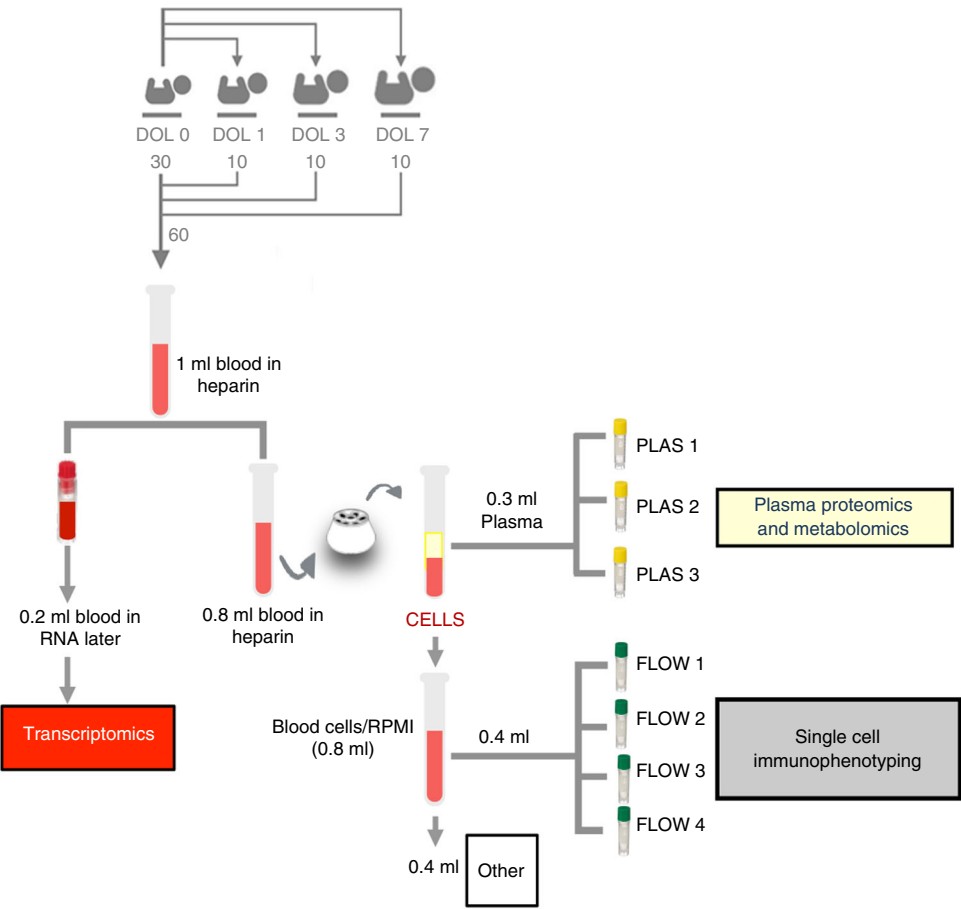

**Fig. 1** Sample processing overview. Thirty newborns were recruited in The Gambia, with each newborn providing a peripheral blood sample on DOL0 and subsets of ten newborns each providing a second peripheral blood sample at either DOL1, 3 or 7, resulting in a total of 60 blood samples. Newborn peripheral venous blood was drawn directly into heparinized collection tubes. Aliquots (200 μl) were removed for transcriptomic analysis. Plasma was then harvested from the remaining whole blood after a spin, and cryopreserved for cytokine, proteomic and metabolomic analyses. The remaining cellular fraction was diluted with phosphate-buffered saline (PBS) to replace the volume of plasma removed, and 100 μl aliquots from this mixture were processed for single-cell immunophenotyping by flow cytometry. With a starting volume of 1 ml, this standard operating protocol still left the cellular fraction contained in 400 μl of starting blood volume that could be used for other analyses. DOL: day of life

Figure 4). Of these, 199 proteins met our criteria, including detection of at least two unique peptides per protein, for further detailed quantification. Substantial between-subject variability was also noted in the plasma proteomic analysis, but again indexing each participant to their baseline enhanced signature detection. This approach indicated a common developmental trajectory over the first week of life, with differences in plasma protein composition compared to DOL0 increasing with increasing age (Fig. 3b). Differentially abundant plasma proteins and their respective pathways are listed in Supplementary Data 6 and Supplementary Methods. At DOL3 vs. DOL0, three pathways were upregulated that center around the complement cascade; at DOL7 vs. DOL0, five additional pathways were upregulated including scavenging heme from plasma and signaling to RAS.

**Metabolomic analysis across first week of life**. While initial analysis revealed substantial between-subject variation, indexing metabolomic data to DOL0 also revealed a steady but dramatic developmental trajectory in the plasma metabolome (Supplementary Figure 5). Few differences in plasma metabolites were identified comparing DOL1 vs. DOL0, but increasing differences were noted when contrasting DOL3 or DOL7 vs. DOL0 (Fig. 3c). Interestingly, metabolomic differences detected across age involved pathways related to plasma steroids and carbohydrate

metabolites, possibly reflecting neurodevelopment, rapid cell proliferation and increased uptake of nutrients in newborns (Supplementary Data 7 and Supplementary Methods)[16].

**Data integration**. Each methodologically- and biologically distinct data type that we examined revealed substantial changes over the first week of life. We thus sought to determine if the observed changes were related to one another across data types, validating consistent age-dependent changes in functional pathways. To minimize the limitations inherent in any single analytical approach, and to detect the most robust signatures, data integration was addressed using three independent strategies. To this end, we employed a novel function-based strategy based on biologically known Molecular Interaction Networks using NetworkAnalyst[17]; an unbiased data-driven multivariate matrix factorization approach using DIABLO (Data Integration Analysis for Biomarker discovery using Latent cOmponents)[18–20]; and the multiscale, multifactorial response network (MMRN) approach that estimates correlations across data types[7,21].

NetworkAnalyst enables the creation of networks based on a framework of known protein−protein interactions (PPI) captured in publicly curated databases (specifically InnateDB/IMeX)[22]. Minimum-connected networks were constructed from seed nodes (i.e. from genes or proteins that changed with age in our dataset),

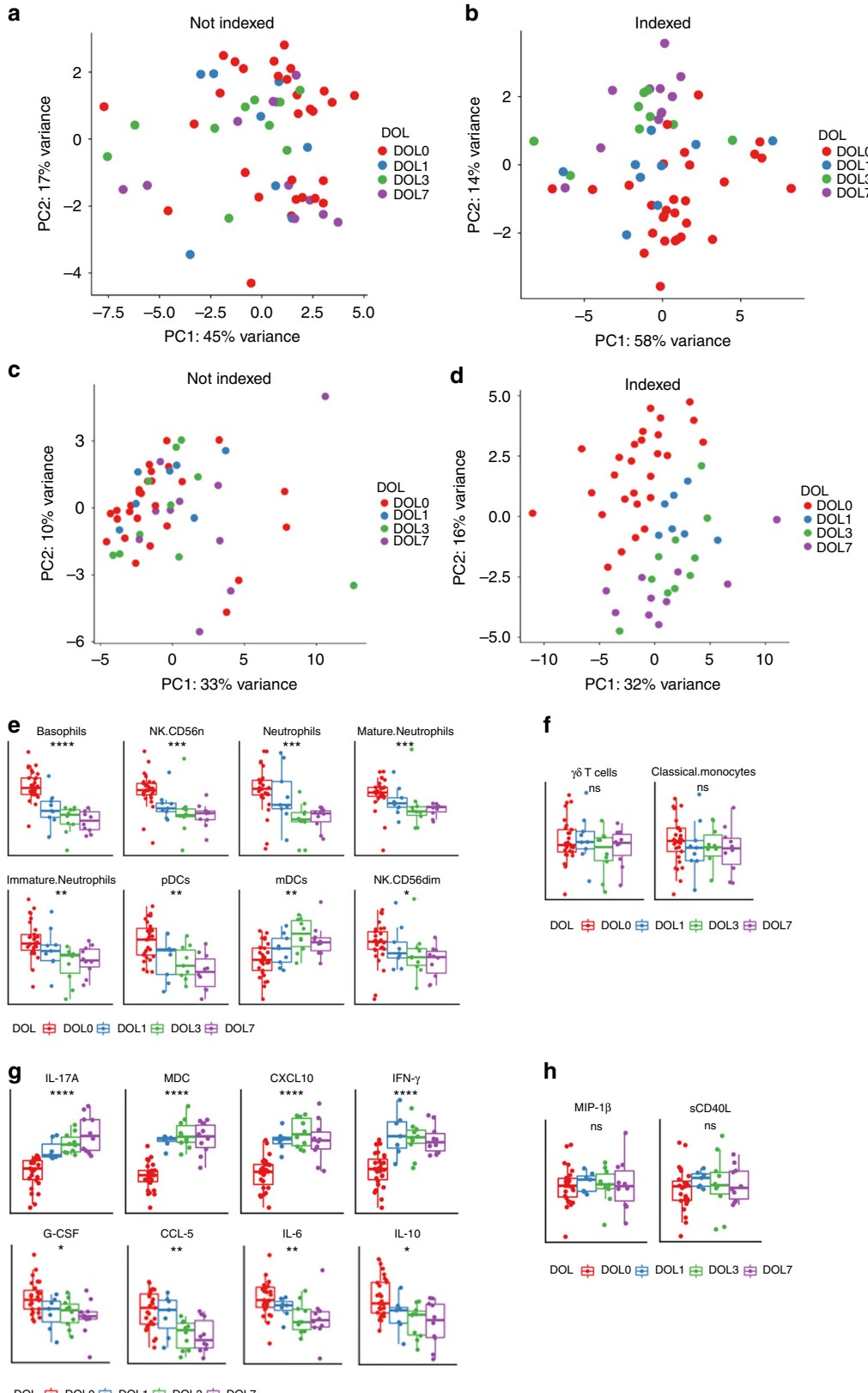

as well as first-order interactors that served to connect the seed nodes with each other. To incorporate metabolomics data, synthetic and degradative metabolic enzymes that would determine the levels of differentially detected metabolites were used as seed nodes in network construction. Overall metabolomics, proteomics and transcriptomics data fit well into a single functional network (Fig. 4), indicating that these techniques reported on different facets of the same biological processes. This PPI-based integration strategy recapitulated many key findings that had been identified for each of the individual data types, confirming our expectation that many but not all findings would be validated across different OMICS datasets (Supplementary

**Fig. 2** Indexing cellular and soluble immune markers revealed developmental progression over the first week of life. **a**, **b** Principal component analysis was used to plot cellular composition (**a**) and plasma cytokines/chemokine concentration (**b**) for each sample; this highlighted the substantial variability between participants and lack of defined clustering by DOL due to higher influence of individual variance over ontogeny. **c**, **d** Accounting for repeat measures from the same individual across different sampling days compared to DOL0 (indexing to DOL0) revealed sample clustering by DOL between samples. **e**, **f** Normalized cell counts showing developmental trajectories for cell populations that significantly changed (**e**) or did not change (**f**) over the first week of life. **g**, **h** Normalized plasma cytokine/chemokine concentrations showing developmental trajectories for cytokines/chemokines that significantly changed (**g**) or did not change (**h**) over the first week of life. Boxplots display medians with lower and upper hinges representing first and third quartiles. Whiskers reach the highest and lowest values, no more than 1.5× interquartile range from the hinge. ****$p ≤ 0.0001$, ***$p ≤ 0.001$, **$p ≤ 0.01$, *$p ≤ 0.05$, ns $p > 0.05$, Kruskal−Wallis test, Benjamini−Hochberg adjusted $p$ values. DOL: day of life

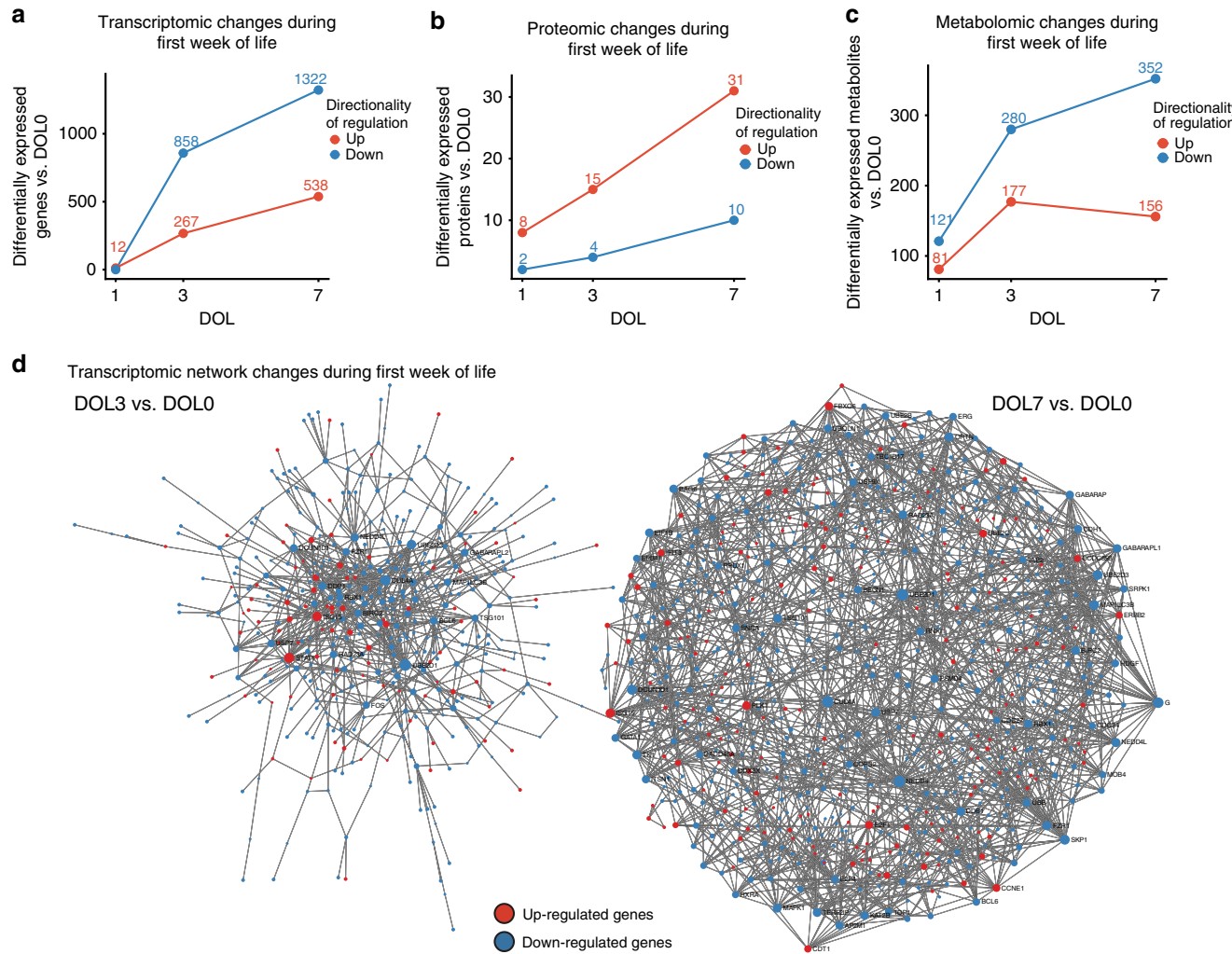

**Fig. 3** Transcriptomic, proteomic, and metabolomic analyses identified a robust trajectory of differentially expressed genes, proteins, and metabolites over the first week of life. **a** Up- and downregulated differentially expressed genes were plotted by DOL (vs. DOL0) and numbers of genes are listed above each point except for downregulated genes at DOL1 vs. DOL0, where the number was zero. **b**, **c** Up- and downregulated differentially expressed proteins and metabolites, respectively, plotted by DOL compared to DOL0, with numbers of differentially expressed proteins or metabolites listed above each point. **d** Zero-order interaction networks for genes differentially expressed at DOL3 vs. DOL0 and DOL7 vs. DOL0. Within networks, upregulated nodes are displayed in red and downregulated nodes in blue. DOL: day of life

Data 8). For example, integrating transcriptomic with proteomic data affirmed the increase in type 1 IFN-related functions and the regulation of complement cascades over the first week of life. This integration also revealed new biological insights not found in any single-data domain analysis, such as changes in cellular replication machinery, creatine metabolism (DOL3), fibrin clotting cascade and signals of increasing adaptive immune and phagosome activity (DOL7).

DIABLO is a multivariate approach to address two of the major concerns faced when integrating multicomponent datasets, namely the complexity of the data, particularly with few samples, each with many observations, and the heterogeneous nature of data measured on different scales and technological platforms[18–20]. DIABLO constructs components (linear combinations of the original data—cells, cytokines, transcripts, proteins, metabolites) that are maximally correlated across any number of input data types with a specified outcome variable (in this case, DOL), while simultaneously performing marker selection[23] to identify a minimal subset of markers associated with this outcome (Supplementary Methods). We created matrices from our five

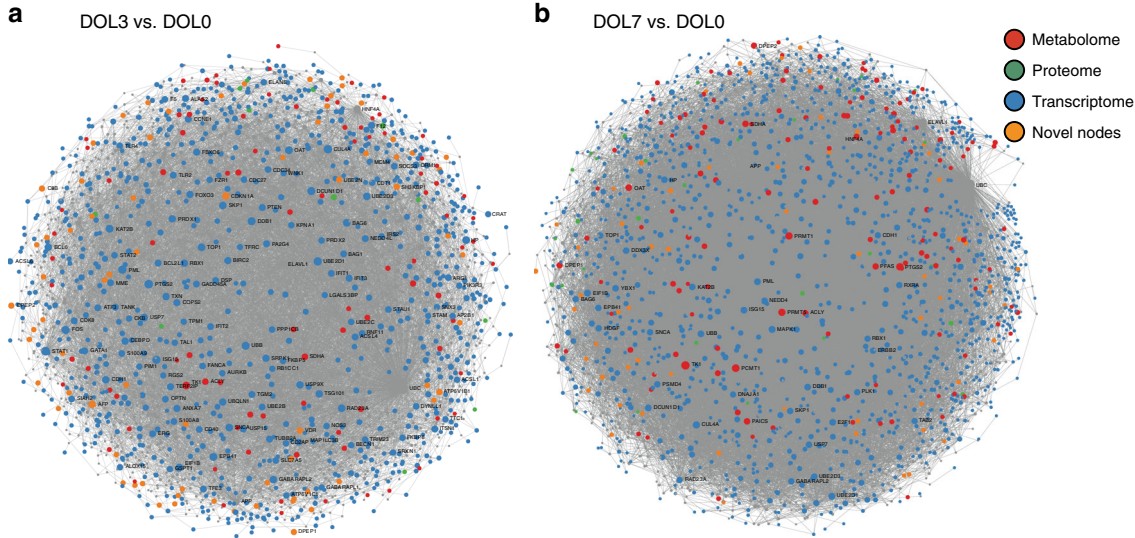

**Fig. 4** Integration of multiple data types via NetworkAnalyst molecular interaction networks provided novel biological insights. Minimum-connected networks for DOL3 vs. DOL0 (**a**) and DOL7 vs. DOL0 (**b**), respectively, containing all three individual data types, where nodes derived from the transcriptome are shown in blue, nodes from the metabolome in red, and nodes from the proteome in green. Novel nodes, which are nodes that only appeared after integrating the three data types but are not present in the individual minimum network, are shown in orange. DOL: day of life

data types as an input to DIABLO to identify major ontogeny-related markers (Fig. 5, Supplementary Figure 6 and Supplementary Data 9). The resulting model discriminated well between DOLs, outperforming nonintegrative approaches (Supplementary Figure 6), with component one of the model separating birth (DOL0) from all other time points, while component two separated DOL1, 3, and 7 from each other. We next investigated the relationship between features selected by DIABLO across data types and visualized the selected features in an integrative network (Supplementary Methods). We compared this integrative network (Fig. 5a, b) to one derived from features identified using an alternative, nonintegrative sparse discriminant analysis approach (Supplementary Methods). The integrative network was more densely connected (global clustering coefficient = 0.91 vs. 0.68) and composed of few, more tightly connected modules (network modularity = 0.26 vs. 0.09), indicating that DIABLO selected features that were discriminant and well correlated across data types, while the nonintegrative approach favored markers that were not well correlated across data types. The two components of the DIABLO model were composed of distinct sets of features (Fig. 5d, blue bars), representing distinct biology (Fig. 5e, blue bars). The first component reflecting DOL0 was composed of markers consistent with interferon and cytokine signaling, among other immune biology (Fig. 5a, Supplementary Methods). The second component, reflecting progression across DOL1–7 had a distinct granulocyte-flavor, as well as a focus on cytokine signaling in the immune system and cellular response to latent infections (Fig. 5a, Supplementary Methods).

MMRNs are a recently published framework for data integration[7]. Using MMRN, we found that associations between data types were strongest at DOL1 and decreased across the first week of life (Supplementary Figure 7), confirming the already noted robust trajectory of development. Most of the significantly correlated clusters were transcriptomic (16/21), but we also identified metabolomic (1/21) and flow cytometry-derived (4/21) clusters associated with DOL (Supplementary Data 10). The stable clusters most significantly associated with DOL were composed of blood transcriptomic modules (BTMs) related to DCs and monocytes including cytokine receptors *CCR1* and *CCR7*, *TLR* and inflammatory signaling, heme biosynthesis,

various B-cell subpopulations, as well as metabolic pathways such as purine metabolism[7].

**Meta-integration**. To assess the similarities across methodologically distinct integration methods, we analyzed the list of nodes associated with DOLs for each integration approach. We then carried out pathway enrichment to determine which biological processes were identified by each method (Supplementary Data 11 and 12). The selected nodes for Molecular Interaction Networks (NetworkAnalyst) were those nodes of minimum-connected networks when differentially abundant features for DOL3 vs. DOL0 and DOL7 vs. DOL0 were used as seed nodes (3195 features). For DIABLO, a first-order molecular interaction network was constructed using as seed nodes the markers derived from model components 1 and 2 (428 features). For MMRN, selected nodes were those that comprised the stable network shown in Supplementary Figure 7 (675 features).

We observed limited overlap at the individual marker level. However, assessing the above networks using the Reactome pathway annotation system (2208 pathways) revealed that 635, 308 and 84 pathways were significantly over-represented in the NetworkAnalyst, DIABLO and MMRN feature lists, respectively (paired *t* test DOL0 vs. DOL1, 3, or 7, Benjamini−Hochberg corrected FDR ≤ 0.05; Supplementary Data 11). Importantly, 249 and 25 pathways were identified by NetworkAnalyst and either DIABLO or MMRN, respectively, and 34 pathways were identified by all three approaches, demonstrating a strong congruence between the outputs and conclusions derived from different analytical/OMICS platforms (Fig. 6a). This degree of overlap was unlikely to occur by chance alone as determined by bootstrapping (*p* value < 0.001; Supplementary Figure 8 and Supplementary Methods). The specific pathways identified by this meta-integration as driving molecular ontogeny over the first week of life were related to interferon signaling, complement cascade and granulocyte function (Supplementary Data 12). Given the striking convergence observed, we next assessed relevant functional interactions between the pathways identified in silico via our meta-integration. To this end, we used NetworkAnalyst and found that selected common features fitted into minimum-connected networks of experimentally validated

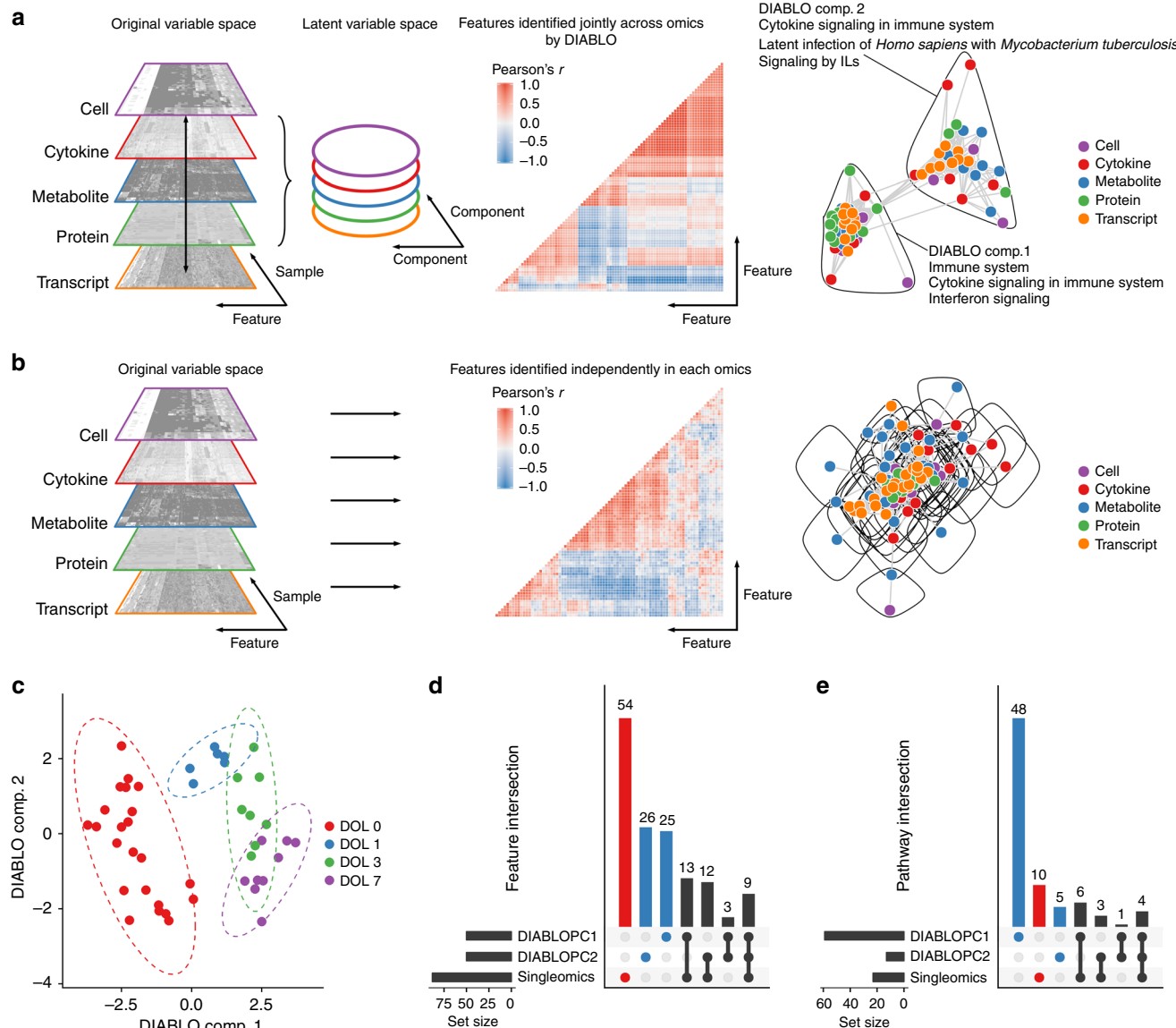

**Fig. 5** DIABLO uncovered biologically relevant features by integrating information across data types. Schematic representation of two contrasting integration approaches using multivariate techniques: **a** shows that DIABLO selects features jointly across data types, resulting in the identification of features with strong associations across data types. Conversely, as shown in **b**, ensembles of multivariate models, constructed independently of each other, result in a selection of features that are poorly associated across data types. This is visualized in correlation heatmaps of the selected features and corresponding networks, with dense subgraphs, or network modules, encircled. In particular, the network modules identified in (**a**) include a number of features selected from all data types. This is not the case in **b**. The minimal set of features selected by DIABLO across data types as shown in **c** could discriminate between DOL and distinct sets of these features separated DOL0 from all other DOLs (DIABLO component 1) and DOL1, 3, and 7 from each other (DIABLO component 2). Features identified by DIABLO (blue bars) were largely distinct from those identified by more traditional single-OMICs multivariate approaches (red bars; overlaps in gray); shown in **d** using an UpSet plot. Moreover, features identified by DIABLO were more strongly enriched for known biological (functional) pathways; shown in **e** using an UpSet plot (blue vs. red bars). Horizontal bars are mapped to the number of elements in each set of features being compared. Vertical bars correspond to the number of elements in the intersections when carrying out various set comparisons. DIABLO: Data Integration Analysis for Biomarker discovery using Latent cOmponents, DOL: day of life

interactions (Supplementary Figure 8), implying functional interconnections. This validation confirmed that our meta-integration approach identified functionally important biological interactions central to neonatal ontogeny.

**Cross-cohort validation.** To validate the generalizability of these integrative models based on data from Gambian (West Africa) newborns, we recruited and characterized an independent newborn cohort (30 participants) from a different region of the world (Papua New Guinea (PNG), Australasia). This validation cohort

was processed according to the same methods as described above (Fig. 1 and Supplementary Figure 9). The outputs for the PNG data demonstrated the same basic trajectory as a function of DOL and showed considerable overlap (e.g. in transcriptomics $p < 10^{-138}$ for the Jaccard index/similarity for DOL3 vs. DOL0 and DOL7 vs. DOL0 in the two cohorts). Overall, integrative multivariate modeling using DIABLO predicted the correct DOL well for these independent cohorts (Fig. 6, Supplementary Figure 10). To quantify the predictive performance of this model, we used the area under the receiver operator characteristics curve (AUROC). Overall predictive accuracy was very high for the transcriptomic

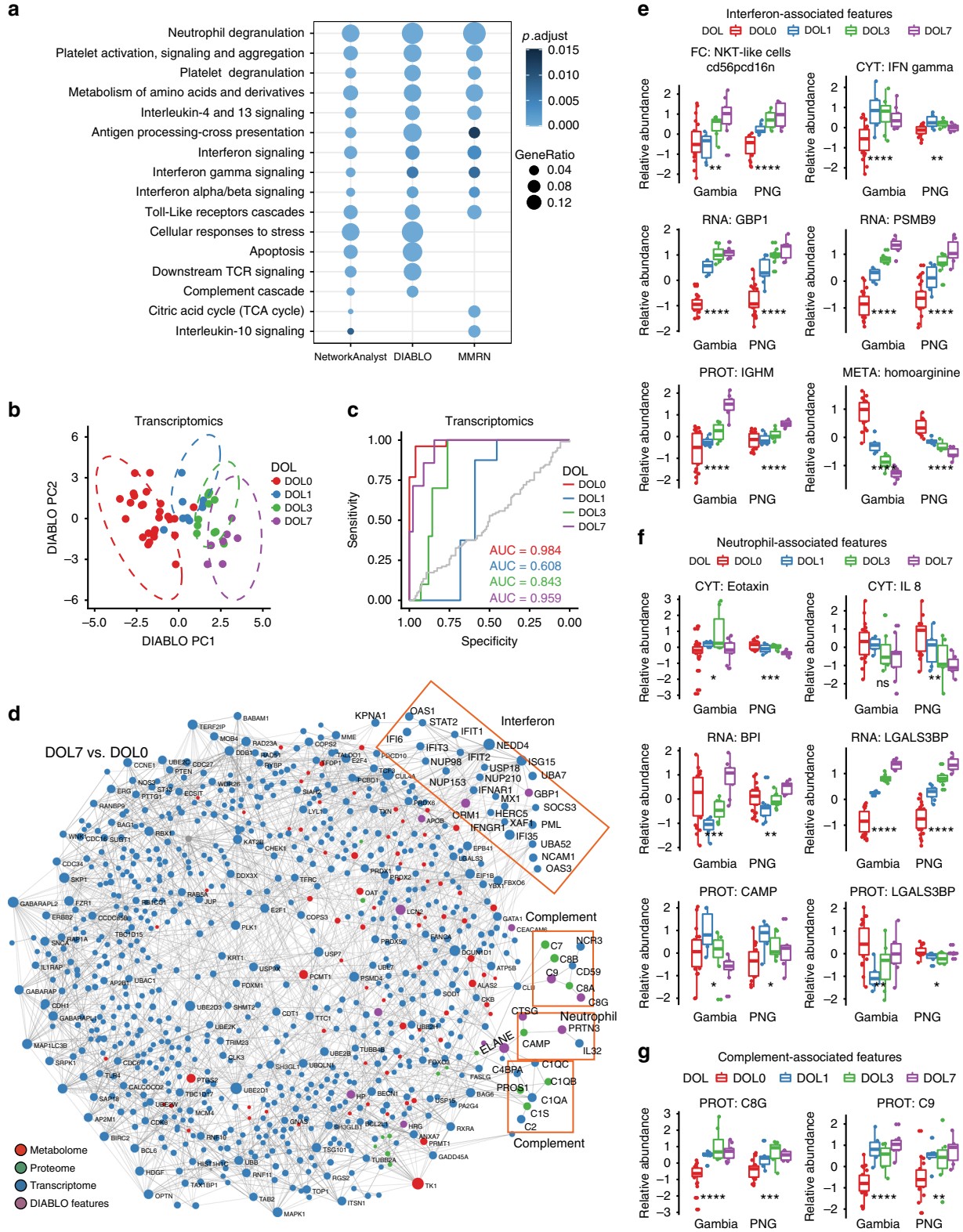

data, with samples from DOL0 and DOL7 separating from all other DOL and near-perfect AUC values (0.98 and 0.95, respectively) while samples from DOL3 or DOL1 were more challenging to classify (0.84 and 0.60, respectively). A functional zero-order network that included key DIABLO-selected features that predict the DOL highlighted in the PNG cohort pathways related to interferon signaling, complement and neutrophil activity (Fig. 6d), as was observed for the Gambian cohort. Thus, we

were able to validate the complex Gambian dataset in an independent and distinct population of newborns.

## Discussion

We present a holistic suite of complementary methods that addressed key hurdles of applying systems biology to the first week of human life by: (i) overcoming limitations in sample volume via an efficient, field-compatible, sample-sparing SOP to

**Fig. 6** Independent validation and data meta-integration of the robust developmental trajectory during the first week of life. Generalizability of the multivariate integrative model (DIABLO) depicted in Fig. 5 based on data from Gambian newborns was evaluated by assessing its ability to classify DOL from OMICs profiles in a new set of validation samples collected from newborns from a second site (Papua New Guinea (PNG)). **a** Pathway enrichments of Molecular Interaction Networks Integration, DIABLO and MMRN identified congruent functional pathways of the first week of life. **b** The dashed line corresponds to the 95% confidence level ellipses for the scores obtained from the Gambia training data. Samples from the PNG site generally resided within the correct ellipse, demonstrating good agreement between actual DOL and DOL as predicted by the model. Similar figures were generated for other OMICs data (Supplementary Figure 10). **c** This agreement was quantified using area under the receiver operator characteristics curve (AUROC) analysis comparing DOL0 (red), 1 (blue), 3 (green), and 7 (purple) individually vs. all other DOLs combined. **d** shows zero-order interaction networks for DOL7 vs. DOL0 containing nodes for transcriptome (blue), proteome (green), metabolome (red), and DIABLO-selected features (purple). Genes involved in the interferon and complement pathways and neutrophil degranulation are highlighted by the orange boxes. **e−g** Relative abundance of a selected subset of markers identified by DIABLO are shown for each DOL for both the Gambian cohort, on which the model was trained, and the validation cohort from PNG. The cells (flow cytometry; FC), plasma cytokines (Luminex assay; CYT) and plasma proteins (mass-spectrometry proteomics; PROT), transcripts (RNA-Seq; RNA), and metabolites (mass-spectrometry metabolomics; META) identified by DIABLO were associated with interferon signaling (**e**), neutrophil recruitment and activation (**f**), and complement pathways (**g**). The differences observed between DOLs in the Gambia cohort were generally replicated in the PNG cohort. Boxplots display medians with lower and upper hinges representing first and third quartiles; whiskers reach the highest and lowest values no more than 1.5× interquartile range from the hinge ****$p \leq 0.0001$, ***$p \leq 0.001$, **$p \leq 0.01$, *$p \leq 0.05$, ns $p > 0.05$, by ANOVA. DIABLO: Data Integration Analysis for Biomarker discovery using Latent cOmponents, DOL: day of life, MMRN: multiscale, multifactorial response network

process peripheral whole blood; (ii) determining the developmental trajectories from each participant's own baseline (indexing), and (iii) reducing the complexity of the dataset by integrating biologically distinct data types generated via a variety of methods (multi-OMICS integration), which enabled validation of signatures of important pathways. The major conclusions were validated in two independent and distinct cohorts of newborns. This approach revealed a dramatic molecular ontogeny evolving over the first week of human life along a common and robust developmental trajectory. Our observation indicates that development over the first week of human life is not immature and random but follows a precise and age-specific path.

This approach was affirmed by its ability to identify known changes in early life, including those related to the composition of hemoglobin mRNA, increases in complement protein *C9* and reductions in plasma steroid metabolites such as pregnanolone/allopregnanolone sulfate and pregnanediol-3-glucoronide[24] (Supplementary Figure 5) that are important for neonatal neurodevelopment[16], supporting the validity of our findings. Moreover, our approach also identified pathways never previously identified as relevant to ontogeny. For example, many pathways upregulated across the first week of life consistently identified by all of our analytical strategies are relevant for host defense but have not before been identified as central to early human development. These pathways included interleukin 1 signaling, Toll-like receptor signaling, NOTCH signaling, the DHX RNA helicase pathway leading to interferon signaling, as well as regulators of the complement cascade[1]. Upregulation of these pathways across the first week of life may represent a heightened defense function for vulnerable newborns who are especially dependent on innate immunity[1]. Data integration also identified novel and surprising but biologically plausible findings regarding ontogeny. For example, prostaglandin-endoperoxide synthase 2 (*PTGS2* or *PGHS2* or *COX-2*) appeared as centrally important in all of our integrative networks, but not in any of the single OMICS data types analyzed individually. *PTGS2* is expressed abundantly in hematopoietic progenitors[25] and is clinically relevant during premature labor and in necrotizing enterocolitis[26,27]. Taken together, our findings outlined a developmental process that can serve as a reference, akin to the stable steady state in adults[7,10,12]. Deviations from this developmental trajectory could potentially identify at-risk individuals prior to disease onset, indicate potential benefit for interventions, or guide the re-establishment of homeostasis once derailed.

Multi-OMICS integration poses an exceptional problem due to the increase in complexity ($p$ data points) relative to the typically achievable sample size ($n$ samples). Here we addressed this $p \gg n$ problem by reducing data complexity via selection of a subset of variables (MMRN, DIABLO)[28], or, for the NetworkAnalyst approach, imposing functionally relevant molecular interaction information[21]. Furthermore, we applied integrative methods focused on extracting aggregated information from each dataset independently of each other, providing high confidence in results that converged on the same set of key molecular features. Overall our approach confirmed that integrating across multiple different biological data types dramatically increased robust biological insights. Technical and analytical improvements, such as implementation of our enhanced plasma proteomics workflow[29], will further increase our understanding of the dynamic molecular changes.

Our integrative approach identified several key pathways as central to ontogeny. These pathways centered around and interconnected interferon signaling, the complement cascade, and neutrophil activity. Each of these have previously been recognized individually as possible contributors to the increase in newborn susceptibility to infection[1], but their age-dependent change had not previously been known to be centrally important to ontogeny over the first week of human life. Importantly, the three key pathways constituting the core of an early life developmental trajectory were readily validated across two independent cohorts of newborns from very different populations, further confirming that the observed early life trajectory is common and predictable and thus could serve as a baseline reference. The finding of a stable trajectory in the first week of life has profound implications: the newborn immune system is still often viewed as immature, which implies a stochastic, unregulated state. However, the existence of the developmental trajectory we discovered as shared by newborns across very different populations in the world strongly argues that early life immune ontogeny is not random but follows a precise and thus possibly purposeful path[1]. Alternatively, the developmental trajectory we observed over the first week of life could be the result of a limited range of response possibilities in early life, as recently espoused in the hypothesis of stereotypic immune system development[4]. Differentiating between these possibilities (a proactive, purposeful path vs. a reactive, restricted response) will be critical for understanding normal development as well as diagnosing, preventing and treating disease in early life[1].

In summary, our study presents an integrative analysis across the broadest range of multivariate datasets published to date covering the first week of human life. Our integration over time (indexing data longitudinally for each newborn) and biological space (multi-OMICS integration) enabled exploration of the dynamic molecular and cellular developmental characteristics of early life. In parallel,

we identified signals of potential physiological importance using a practical sample size. Coupled with our field-tested SOP for sample-sparing preanalytical processing, this strategy overcomes key challenges in applying systems biology to neonates. Our data revealed a compelling biological narrative, where out of the apparently noisy age-dependent changes across the first week of life, a consistent developmental trajectory emerged. The interactive contributors to this trajectory relate to cell autonomous as well as innate immune functions, such as interferon-driven signaling, complement cascades, and neutrophil function[1]. Our approach and the resulting observations will serve as a crucial backdrop for future studies that characterize the impact of a broad array of factors, including genetics, epigenetics, maternal influences, microbiota, diet, and disease, as well as chemical and biomedical interventions such as antibiotics and vaccination.

## Methods

**Overview**. A workflow diagram is provided in Supplementary Figure 1 that outlines the precise sample processing steps as well as the statistical and bioinformatic analyses steps applied to the data.

**Peripheral blood processing**. Thirty healthy, term newborns were enrolled at each of the Medical Research Council (MRC) Unit The Gambia and at the Institute for Medical Research (IMR) in Goroka, Papua New Guinea in accordance with a local Ethics Committee-approved protocol (MRC SCC 1436 and IMR IRB#1515 and MRAC #16.14). Informed consent was obtained from all human participants. The initial numbers of participants recruited for the Gambian cohort were based on data outputs rather than formal power calculations. This effect size from the Gambian dataset, however, allowed us to estimate the sample size needed for our subsequently enrolled PNG validation cohort. Specifically, using the approach described by Liu et al.[30], the mean fold-change and standard deviation were estimated in the Gambia transcriptomic data (using the data in Supplementary Data 5–12), and, controlling the FDR at 5%, under different assumptions for the proportion of non-DE features (0.8, 0.9), based on what was observed in the Gambian cohort (~10% of transcripts DE at DOL3 and ~15% of transcripts DE at DOL7), we estimated that with $n = 10$, we would have >80% power, even when assuming a very low proportion of DE features (10%).

Following informed consent, mothers were screened for HIV-I and -II and Hepatitis B with positivity for either virus representing an exclusion criterion. Inclusion criteria were a healthy appearing infant as determined by physical examination, born by vaginal delivery at gestational age of >36 weeks, 5-min Apgar scores > 8, and a birth weight of >2.5 kilograms. Peripheral blood samples were obtained from all infants on the day of birth (DOL0) and then again either at DOL1, DOL3 or DOL7, in order to reduce venipuncture to a maximum of twice in the first week of life. Peripheral venous blood was drawn from infants via sterile venipuncture directly into heparinized collection tubes (Becton Dickinson (BD) Biosciences; San Jose, CA, USA). Aliquots (200 μl) were immediately placed in RNA-later (Ambion ThermoFisher; Waltham, MA, USA) with the remaining blood kept in the collection tubes at room temperature until further processing within 4 h. All samples were processed as described below and subsequently shipped to collaborating laboratories on dry ice, under temperature controlled and monitored conditions (World Courier; New Hyde Park, NY, USA).

**Indexing**. The peripheral blood of each participant was profiled twice over their first week of life, i.e. at DOL0 and additionally at either DOL1, 3, or 7, as we sought to identify variables that differed between the baseline and latter time points across all participants. Obtaining a baseline DOL0 sample for each participant enabled indexing as described in Methods. For univariate analyses, we considered (indexed) paired differences implicitly using paired statistical tests (e.g. paired *t* test); for multivariate analyses, participant sample pairing was addressed explicitly by transforming the data beforehand using a multilevel approach to separate the between- and within-subject variation[31]. We use the term indexing to refer to each of these treatments, as appropriate, throughout the text.

While the limited number of samples available did not allow us to consider additional potential confounders, such as when feeding started, first passing of meconium or birth weight, our approach of indexing data longitudinally for each newborn allowed us to look for consistent differences between each DOL. While other unforeseen confounders may exist, they did not obscure the strong developmental patterns in the data observed during the first week of life across two very different cohorts.

**Immune phenotyping**. Whole blood was centrifuged on site at $500 \times g$ for 10 min at room temp and plasma harvested and stored at −80 °C for later analysis of plasma cytokines, proteins and metabolites. The amount of plasma removed from the whole blood after centrifugation was subsequently replaced with RPMI. For assessment of cellular composition by FCM, ethylenediaminetetraacetic acid

(EDTA) (0.2 mM final concentration) was added to the whole blood/RPMI mixture to ensure adherent cells were not lost. In parallel, cells were stained with fixable viability dye at 4 °C for 15 min prior to red blood cell lysis followed by storage at −80 °C in Smart Tube reagents (Smart Tube Inc.; San Carlos, CA, USA). At the immunophenotyping laboratory samples were thawed, washed in staining buffer (PBSAN; 0.5% bovine serum albumin (BSA), 0.1% sodium azide in PBS) and stained on ice in PBSAN with a cocktail of anchor markers to determine frequency of cell populations contained in peripheral blood (for list of cell types and anchor markers, see Supplementary Figure 2; for list of clone/ fluorochrome combination, see Supplementary Note 1). Flow cytometric analysis employed a custom-built LSRII (for machine settings and compensation settings, see Supplementary Note 1) [32]. Our gating strategy is shown in Supplementary Figure 2. FCM data were analyzed in an automated fashion using R/Bioconductor packages (Supplementary Figure 2). Specifically, flowCore supported the analysis in single files according to the Flow Cytometry Standard (FCS), providing the infrastructure to support subsetting of data, data transformations and gating[33]. Cell population identification was then conducted using flowDensity, a supervised gating tool, that was customized to provide threshold calculations designed for each cell subset based on expert knowledge of hierarchical gating order and one-dimensional density estimation[34]. Lastly, flowType/RchyOptimyx identified cell populations that correlated with outcome, in this case DOL at the time of blood draw[35]. flowType uses partitioning of cells, either manually or by clustering, into positive or negative for each marker to enumerate all cell types in a sample. RchyOptimyx measures the importance of these cell types by correlating their abundance to external outcomes, such as DOL, and distills the identified phenotypes to their simplest possible form.

Plasma (25 μl) was used to measure cytokine concentrations using a custom-designed multi-analyte Cytokine Human Magnetic Panel bead array, (Invitrogen/Life Technologies; Carlsbad, CA) consisting of *CCL2, CCL3, CCL5, CXCL8, CXCL10, GM-CSF, IFN-a2, IL-10, IL-12p40, IL-12p70, IL-1β, IL-6*, and *TNFα*. Results were obtained with a Flexmap 3D system with Luminex xPONENT software version 4.2 (both from Luminex Corp.; Austin, TX, USA). Cytokine concentrations were determined using Milliplex Analyst software (version 3.5.5.0, Millipore).

For flow cytometric as well as Luminex raw values were normalized with a 1+ Log2-transformation. WithinVariation matrices were computed for each data matrix using the WithinVariation function in R package mixOmics version 6.1.2. The Kruskal−Wallis test (kruska.test in base R) was used to determine differentially regulated features within each data type, using the WithinVariation values for each feature. These *p* values were adjusted for each data type separately using the Benjamini−Hochberg method (*p*.adjust function, base R). Features were considered statistically different by day of life from DOL0 if their adjusted *p* values were below 0.05. All analyses were performed in R version 3.3.2 (2016-10-31).

**RNA-Seq**. Total RNA was extracted from each sample using the RiboPure RNA purification kit (Ambion ThermoFisher; Waltham, MA, USA) following the manufacturer's protocol. Quantification and quality assessment of total RNA was performed using an Agilent 2100 Bioanalyzer (Santa Clara, CA, USA). Poly-adenylated RNA was captured using the NEBNext Poly(A) mRNA Magnetic Isolation Module (catalog no.: E7409L, NEB; Ipswich, MA, USA). Strand-specific cDNA libraries were generated from poly-adenylated RNA using the KAPA Stranded RNA-Seq Library Preparation Kit (cat. no.: 07277253001, Roche; Basel, Switzerland). All cDNA libraries were prepared at the same time and sequenced on the HiSeq 2500 (Illumina; San Diego, CA, USA), using one Rapid v2 and two lanes of High Output single-read run of 100 bp-long sequence reads (+ adapter/index sequences). Sequence quality was assessed using FastQC v0.11.5 and MultiQC v0.8. dev0 [36]. The FASTQ sequence reads were aligned to the hg38 human genome (Ensembl GRCh38.86) using STAR v2.5 and mapped to Ensembl GRCh38 transcripts[37]. Read-counts were generated using htseq-count (HTSeq 0.6.1p1)[38]. All data processing and subsequent differential gene expression analyses were performed using R version 3.3.0 and DESeq2 version 1.14.1 [39]. Genes with very low counts (with less than ten counts in eight or more samples, or the smallest number of biological replicates within each treatment group) and globin transcripts were prefiltered and removed in silico. Differentially expressed genes were identified using paired analysis with the Wald statistics test and filtering for any genes that showed twofold change and adjusted *p* value < 0.05 (cut-off at 5% FDR) as the threshold. Functional discovery of pathway enrichment and network analyses was performed using SIGORA v2.0.1 and NetworkAnalyst, respectively[17,40]. To test whether changes in cell composition could account for the observed changes in gene expression, we used DESeq2 (with default parameters) to fit two models, one including subject and DOL and the other model including the additional covariate of cell composition. To address the collinearity of this compositional data, we used PCA, summarizing the cell proportions (flow cytometry) to five principle components (PCs, accounting for 95% of the variance observed). We compared the estimated effect sizes for the DOL term between these two models (Pearson correlation) across all genes and found them to be highly correlated ($p < 10^{-50}$), indicating that the observed changes in gene expression could not be (fully) explained by changes in the underlying cell composition.

**Plasma proteomics**. Plasma samples were prepared for proteome analysis using an in-house (Boston Children's Hospital; Steen laboratory)-developed MStern blotting sample processing and trypsinization protocol[41], which was adapted for

plasma samples[42]. To this end, 5 µl plasma was first diluted in 100 µl sample buffer (8 M urea in TRIS-HCl, pH 8.5). Protein disulfide bonds were then reduced with dithiothreitol (10 mM final concentration) for 30 min and alkylated with iodoacetamide (50 mM final concentration) for 30 min in sample buffer. Three µl (approximately 10 µg) of this protein solution was then transferred to a 96-well plate with a polyvinylidene fluoride (PVDF) membrane at the bottom. Protein digestion was performed with sequencing-grade modified trypsin (V5111, Promega; Madison, WI, USA) at a nominal protease to protein ratio of 1:25 w/w. After incubation for 2 h at 37 °C, the peptides were eluted from the PVDF membrane, and concentrated to dryness in a vacuum centrifuge. To monitor retention time stability and system performance, iRT peptides (Biognosys; Schlieren, Switzerland) were spiked into all samples. Samples were analyzed using a nanoLC system (Eksigent; Dublin, CA) equipped with a LCchip system (cHiPLC nanoflex, Eksigent) coupled online to a Q Exactive mass spectrometer (Thermo Scientific; Bremen, Germany). From each sample, 0.2 µg peptide material was separated using a linear gradient from 93% solvent A (0.1% formic acid in water), 7% solvent B (0.1% formic acid in acetonitrile) which was increased to 32% solvent B over 60 min. The mass spectrometer was operated in data-dependent mode, selecting up to 12 of the most intense precursors for fragmentation from each precursor scan. Label-free protein quantitation analysis employed MaxQuant 1.5.3.30 [43]. Raw-data were downloaded and used to build a matching library and searched against the UniProt Human Reference Proteome as described. Standard search settings were employed with the following modifications: Max missed cleavage 3; variable modifications Deamidation (NQ) and Oxidation (M)[44]. A revert decoy search strategy was employed to filter all proteins and peptides to <1% FDR. The list of proteins was further processed in Perseus 1.5.5.3, log2-transformed, and proteins with less than two peptides (razor) were filtered out as described[45]. The samples were grouped according to DOL, and proteins that were not quantifiable in at least five of the samples in any day were filtered out. Remaining missing values were imputed using numbers drawn from a normal distribution with the standard parameters in Perseus (width 0.3, downshift 1.8) to simulate signals from low abundant proteins The R-script ComBat was used to correct for batch effects for samples run on different LC-MS columns[46]. Proteins with a statistically significant change of abundance between different DOL were identified by paired two-sample $t$ test. To correct for multiple hypothesis testing, permutation-based false positive control was applied using standard parameters in Perseus (FDR = 0.05, s0 = 0.1) [47]. Significant proteins were further analyzed using SIGORA[40] with the Reactome[48] gene annotation system and DAVID[49].

**Plasma metabolomics**. Plasma samples were run using the nontargeted metabolomics platform of Metabolon Inc. (Durham, NC, USA). Samples were extracted and prepared using Metabolon's solvent extraction method[50] and run on four independent platforms: reverse-phase/UPLC-MS/MS with positive ion mode electrospray ionization (ESI), reverse-phase/UPLC-MS/MS with negative ion mode ESI, HILIC/UPLC-MS/MS with negative ion mode ESI, and a backup using Waters ACQUITY ultra-performance liquid chromatography (UPLC) and a Thermo Scientific Q-Exactive high resolution/accurate mass spectrometer interfaced with a heated electrospray ionization (HESI-II) source and Orbitrap mass analyzer operated at 35,000 mass resolution. Biochemical identifications of metabolites included the following three criteria: retention index (RI) within a narrow RI window of the proposed identification, accurate mass match to the library ± 10 ppm, and MS/MS forward and reverse scores between the experimental data and authentic standards. The MS/MS scores were based on a comparison of the ions present in the experimental spectrum to the ions present in the library spectrum. Similarities between these molecules were based on one of these factors and the use of all three data points can be utilized to distinguish and differentiate biochemicals. The metabolites were confirmed by comparing their mass spectra and chromatographic retention times with more than 3300 commercially available reference standards. All identified metabolites were categorized to level 1 metabolites according to reporting standards set by the Chemical Analysis Working Group of the Metabolomics Standards Initiative and have appropriate orthogonal analytical techniques applied to the metabolite of interest and to a chemical reference standard. These identified metabolites included corresponding accurate mass via MS with retention index, chemical and composition ID and accurate mass matched. About 75% of the metabolites identified also have Pubchem ID associated with it. MS/MS fragment ion analysis process was performed using peak-matching algorithms (Waters MassLynx v4.1, Waters corp.; Milford, USA) and quantified using an area-under-the-curve in-house algorithm. Metabolite features where all values are missing/undetectable, features with a signal to noise ratio <10 and those with just a single value in the whole dataset were excluded from the samples. All remaining missing values were imputed with half the minimum value of peak intensity for that feature. Features with interquartile range of zero were excluded to ensure reproducibility. All metabolite features were then log-transformed for normalization, pareto-scaled to reduce variation in fold change differences and for robust size effect comparability across samples. Significant metabolite between DOL were identified using $t$ test between paired samples corrected for multiple hypothesis testing (threshold of adjusted $p$ value < 0.05) using MetaboAnalyst 4.0. Hypergeometric test enrichment of metabolite sets were matched against KEGG

Pathway[51], The Human Metabolome Database Version 3.6 (HMDB)[52] and Metabolync Pathway Analysis using the Cytoscape plugin[53].

**Data integration**. Data obtained via the immune phenotyping (cellular composition and plasma cytokines), transcriptomic, proteomic and metabolomic methods was integrated to identify correlations of signatures across these methodologically- and biologically distinct datasets, since convergence of signatures across such diverse biological domains provides an independent assessment and approaches functional validation[7]. In addition, we assessed whether we could derive any novel biological information via the integration of multiple OMICS data types that was not revealed in a single dataset alone. To cross-validate results we employed three data integration platforms, each applying a different analytical strategy; these independent but complementary data-driven vs. knowledge/network-driven strategies, pursued in parallel, decreased the chance of false discoveries. Specifically, data integration strategies included: (i) a novel method for integrating multiple OMICS data types into known protein−protein interaction networks (Molecular Interaction Network) using NetworkAnalyst that provided context based on annotated molecular interactions[17]; (ii) a new approach to identify the underlying key drivers of ontogeny using specialized multivariate methods capable of identifying relevant features from high-dimensional datasets, namely sparse generalized canonical correlation discriminant analysis via the Data Integration Analysis for Biomarker discovery using Latent cOmponent (DIABLO) framework, which is part of the mixOmics R package;[18,19,54] and (iii) the recently published MMRN, querying the informatically derived correlations within a network for statistically significant features[7].

**Molecular Interaction Network**. NetworkAnalyst integrates data using protein −protein interactions as a biological framework. Metabolomic data required pre-processing in order to integrate into a network with transcriptomic and proteomic data, as metabolites must be associated with proteins that are involved in their creation and/or degradation. To identify such proteins, the following steps were taken: metabolites were mapped to their directly interacting enzymes based on their corresponding HMDB IDs via the MetaCyc module in MetaBridge[55,56]. To construct networks for each OMICS type, the list of proteins derived from the metabolite data processing (above), and lists of differentially expressed genes or proteins identified by transcriptomic and proteomic data were submitted to NetworkAnalyst to produce a zero-order or a minimum-connected network, depending on the size of the dataset[17]. Networks consist of seed nodes (proteins/ genes that were used as input to build the network) and edges (links that join the nodes together and are indicative of a molecular interaction between nodes).

Minimum-connected integrated networks were constructed using pairwise or all three OMICS lists of differentially present genes/proteins on each later DOL vs. DOL0 for transcriptomic, proteomic and metabolomic data. Novel nodes in each network were identified as nodes that were not in either set of differentially expressed genes/proteins nor found in minimum-connected networks constructed from each single data type. Our focus was on the identification of novel nodes as emergent information that can be derived from a biological network. Node lists were then downloaded to identify median degree of connectivity for nodes of each data type and to identify novel nodes stemming from data integration. Transcriptomes were integrated via their respective encoded gene products. Pathway enrichment analysis was performed using SIGORA R package[40] with the Reactome ontology system on the node lists from each minimum-connected network built during the novel node identification process outlined above. For each data type alone (transcriptomics, proteomics and metabolomics), the number of unique pathways identified were counted as novel pathways. For each pairwise combination of data types, previously identified pathways that had been identified in either data type alone were subtracted. For the three-way combination all unique data type pathways were subtracted.

**DIABLO**. Prior to integration with DIABLO, data were transformed. Specifically, cell proportions from the flow cytometry were normalized to total cell counts; the resulting relative cell proportions were then transformed using centered log-ratios[57]. Normalized transcriptomic, proteomic, cytokine, and metabolomic data were log-transformed. We further decomposed the within-subject from the between-subject variance in the datasets to account for repeated measures[58]. This is analogous to normalizing all samples to their DOL0 time point. Finally, a broad, unsupervised, variance-based filter was applied to the transcriptomic data, retaining the 50% most variable features, also termed markers in the text. To integrate across data types, we applied sparse generalized canonical correlation discriminant analysis via the DIABLO framework, part of the mixOmics package[18,19,59]. DIABLO constructs components across any number of input matrices, maximizing their covariance with each other and a given response variable (in this case, DOL), while simultaneously performing feature selection[23]. Importantly, DIABLO identifies key drivers associated with the response variable of interest across all input data matrices jointly. Cross-validation (20 × 5-fold) was used to determine the optimal model hyperparameters (number of components, number of features per component), as well as to provide an estimate of the ability of the model to generalize to new data. Selected model features, i.e. key correlates of change across DOL, were subjected to pathway over-representation analysis against the Reactome pathways database (obtained via MSigDB[60]) and BTM[21] annotated

gene set libraries, using the hypergeometric test, either separately per component, or after enrichment to include their first-degree neighbors subgraph in the protein−protein interaction data[17]. Obtained *p* values were adjusted for multiple comparisons using the Benjamini−Hochberg procedure[61].

**Multiscale, multifactorial response network**. Examples of systematic integration across multiple large OMICS datasets remain rare. Recently, Li et al.[7] used MMRNs to study the immune response to varicella zoster vaccine across four OMICS datasets. We wanted to compare insights generated by our chosen approaches to theirs and, to that end, created a multiscale interaction network as they described, with the following important modifications. First, instead of summarizing the higher dimensional data blocks (transcriptomics, metabolomics) to either gene modules or metabolic pathways ((BTM)[62]; Metabolon Inc. annotation, respectively) using a simple average, we used Eigen-gene summarization[63], a weighted average based on the first principal component of the data, in order to maximize the variance explained. Second, clusters were identified at each DOL using Euclidean distance and Ward's method as described[64]. The optimal number of clusters was determined using the elbow criterion[65]. Stable clusters were identified by comparing cluster membership using the Szymkiewicz−Simpson coefficient[66]. Finally, cluster association with DOL was assessed using the Correlation Adjusted MEan RAnk (Camera) gene set test[67].

**Meta-integration to identify convergence across data integration strategies**. We performed meta-integration to address whether: (1) DIABLO-selected features formed a functional network as defined by known protein−protein interactions, and (2) if each of the integration methodology, namely the Molecular Interaction Networks integration, DIABLO and MMRN, would identify similar biological signals across the first week of life. To assess whether DIABLO-selected features formed a functional protein−protein interaction network, 20 features each from transcriptome, proteome, and metabolome data were used to create a minimum-connected network. Metabolites were mapped to their directly interacting enzymes based on their corresponding HMDB IDs via the MetaCyc module in Meta-Bridge[55,56] as above, and used as seed nodes for NetworkAnalyst.

The various data integration approaches resulted in outputs that were not directly comparable. To enable direct comparisons, we simplified their outputs to lists of features of interest. Molecular Interaction Networks integration: list of features that made up a minimum-connected network using nodes from differentially abundant features (transcripts, proteins and metabolites) for DOL3 vs. DOL0 and DOL7 vs. DOL0 (3195 features). DIABLO: a first-order connected network including seed nodes of component 1 and 2 features of the DIABLO model using NetworkAnalyst (428 features). MMRN: features were selected from the small stable interconnected node-network of transcriptomic and metabolomics features shown in Supplementary Figure 7 (675 features). Pathway enrichment of 3195 (Molecular Interaction Networks integration), 428 (DIABLO) and 675 features (MMRN) was done using ReactomePA with the Reactome annotation and the comparison of enriched pathways was performed using clusterProfiler[68,69].

## Data availability

All data are publicly available. The transcriptomics data presented in this publication were submitted to the NCBI Gene Expression Omnibus under accession numbers GSE111404 and GSE123070. All other datasets including immune phenotyping, Luminex, metabolomics and proteomics data were archived on ImmPort (https://immport.niaid.nih.gov/home) under accession numbers SDY1256 and SDY1412.

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

## Acknowledgements

We would like to thank all the participants and their parents for their time and will-ingness to support this study. Research reported in this publication was supported by the National Institute of Allergy and Infectious Diseases of the National Institutes of Health as part of the Human Immunology Project consortium under 5U19AI118608-02. The content is solely the responsibility of the authors and does not necessarily represent the official views of the National Institutes of Health. T.R.K.'s laboratory is supported by a Michael Smith Foundation for Health Research Career Investigator Award. O.L.'s laboratory is supported by the following U.S. National Institutes of Health (NIH)/National Institutes of Allergy and Infectious Diseases (NIAID) awards: Molecular Mechanisms of Combination Adjuvants (1U01AI124284-01), Adjuvant Discovery Pro-gram Contract No. HHSN272201400052C and Human Immunology Project Consortium (U19AI118608) as well as an internal Boston Children's Hospital award to the Precision Vaccines Program. B.K. is supported by grants from the MRC/UKRI (MC_UP_A900/1122, MC_UP_A900/115, MR/R005990/1), and the additional field team and laboratory staff at the MRC Unit in The Gambia. Recruitment of the cohort of newborns in Papua New Guinea was funded by seed funding awarded to A.H.J.v.d.B from the Wesfarmers Centre of Vaccines and Infectious Diseases, Telethon Kids Institute. The work in R.E.W. H.'s lab was initially supported by the Canadian Institutes for Health Research grant # FDN-154287 and he holds a Canada Research Chair in Health and Genomics and a UBC Killam Professorship. R.R.B.'s laboratory is supported by an award from Natural Sciences and Engineering Research Council of Canada. The Lundbeck Foundation (R181-2014-3372), The Carlsberg Foundation (CF14-0561), and A.P. Møller Foundation are acknowledged for grants enabling T.B.B.'s work. K.-A.L.C. is supported in part by the National Health and Medical Research Council (NHMRC) Career Development fel-lowship (GNT1087415). We gratefully acknowledge the support from Drs. Gary Fleisher, Michael Wessels and Ken Kraft as well as Maria Crenshaw, Mark Liu, Kerry McEnaney and Diana Vo (all BCH); Susan Farmer, Manish Sadarangani, Aaron Liu, Gordean Bjornson (all UBC). The Expanded Program on Immunization Consortium (EPIC) contributed collectively to this study. EPIC is an association of academic centers part-nering to conduct systems biology studies in newborns and infants, comprised of the investigators listed above at Boston Children's Hospital (BCH), University of British Columbia (UBC), Medical Research Council Unit The Gambia (MRCG), Université libre de Bruxelles, Telethon Kids Institute and University of Western Australia, and the Papua New Guinea Institute for Medical Research (PNG-IMR).

## Author contributions

Project and core leads were: O.L., T.R.K., H.S., R.E.W.H., S.J.T., A.O., A.H.J.v.d.B., and B. K. Project activities were coordinated by the Administrative Core (O.L., J.D.-A., D.V., K. K.S; BCH; T.R.K., R.B.-O.; UBC). All infants were enrolled at the MRCG (O.I., B.K.) and PNG-IMR (W.S.P., A.H.J.v.d.B.), where biospecimens were collected, processed and shipped (M.C., A.D., J.N.-J., J.N., E.R., R.F., G.S., G.M., J.P.M., W.K.), to the Clinical Core (K.K.S., O.L., S.v.H.; BCH) for further processing. Data generation and analyses were conducted at centers specializing in RNASeq (A.H.L., E.E.G., R.F., R.E.W.H.; UBC), proteomics (T.B.B., H.S.; BCH), cytokine/chemokine analysis (S.v.H, O.L.; BCH; N.A.; D. H.; UBC), flow cytometry (T.R.K., N.A., R.B.-O., C.B., D.J.H.; M.M., R.R.B.; UBC), with metabolomics data provided to O.L. and J.D.-A. (BCH) on a fee-for service basis (Metabolon; Durham, NC). Data underwent QC and QA and were released via the Data Management Core (J.D.-A., S.M.V., K.M., A.O.; BCH; C.P.S.; S.J.T.; UBC & PROOF Centre; R.E.W.H., R.B.-O.; UBC). Writing team: A.H.L., C.P.S., R.B.-O., N.A., T.B.B., J. D.-A., O.L., and T.R.K.; development of the clinical protocol: A.H.J.v.d.B., B.K., O.I., O.L., T.R.K., P.C.R., and W.S.P.; development of the protocol for fractionating small blood samples: G.S.-S., M.A.P., N.A., O.L., R.B.-O., S.v.H., and T.R.K., for RNASeq: A.H.L., E.E. G., R.E.W.H.; proteomics: T.B.B., H.S.; metabolomics: J.D.-A., M.A.P.; flow cytometry and automated flow data analyses: R.B.-O., D.H., D.J.H., R.R.B., M.M., T.R.K., cytokine/chemokine multiplexing: S.v.H; data integration: C.P.S., E.E.G., J.D.-A., A.H.L., S.J.H., A. S., S.J.T., and R.E.W.H. K.-A.L.C. and D.F. contributed to the conceptualization and writing of the bioinformatic analyses. A.A. assisted in interpretation of neonatal mole-cular ontogeny. A.M. assisted with study design. K.K. provided expert input on opti-mizing the structure and function of the EPIC organization. S.M.V. served as the Precision Vaccines Program Data Coordinator for deposition of deidentified study data to NIH's ImmPort and Gene Expression Omnibus (GEO) public repository website. D.V. provide project coordination via the Precision Vaccines Program. All authors reviewed and approved the final version of the manuscript.

## Additional information

**Competing interests:** O.L. is a named inventor on patents regarding bactericidal/permeability increasing protein (BPI), including "Therapeutic uses of BPI protein products in BPI-deficient humans" (WO2000059531A3) and "BPI and its congeners as radiation mitigators and radiation protectors" (WO2012138839A1). R.R.B. has ownership interest in Cytapex Bioinformatics Inc. The remaining authors declare no competing interests.

[1]Department of Microbiology & Immunology, University of British Columbia, 1365-2350 Health Sciences Mall, Vancouver, BC V6T 1Z3, Canada. [2]PROOF Centre of Excellence, 10th Floor, 1190 Hornby Street, Vancouver, BC V6Z 2K5, Canada. [3]Department of Experimental Medicine, University of British Columbia, 2775 Laurel Street, 10th Floor, Room 10117, Vancouver, BC V5Z 1M9, Canada. [4]Department of Pathology, Boston Children's Hospital, BCH 3108, 300 Longwood Ave, Boston, MA 02115, USA. [5]Precision Vaccines Program, Division of Infectious Diseases, Boston Children's Hospital, 300 Longwood Ave, BCH 3104, Boston, MA 02115, USA. [6]Harvard Medical School, 25 Shattuck Street, Boston, MA 02115, USA. [7]Vaccines & Immunity Theme, Medical Research Council Unit The Gambia at the London School of Hygiene and Tropical Medicine, Atlantic Boulevard P.O. Box 273 Banjul, Gambia. [8]Center for International Health, Medical Center of the University of Munich (LMU), Munich, Germany. [9]Department of Pediatrics, BC Children's Hospital, University of British Columbia, Rm 2D19, 4480 Oak Street, Vancouver, BC V6H 3V4, Canada. [10]Papua New Guinea Institute of Medical Research, Homate Street, Goroka, Eastern Highlands Province, Papua New Guinea. [11]Statistical Genomics, School of Mathematics and Statistics, Melbourne Integrative Genomics, Centre for Systems Genomics, The University of Melbourne, Building 184 Ground Floor, Parkville, VIC 3010, Australia. [12]Graduate Program in Bioinformatics, BCCA, 100−570 West 7th Avenue, Vancouver, BC V5Z 4S6, Canada. [13]Division of Paediatrics, School of Medicine, University of Western Australia, 35 Stirling Highway, Nedlands, WA 6009, Australia. [14]Wesfarmers Centre of Vaccines and Infectious Diseases, Telethon Kids Institute, University of Western Australia Perth, 15 Hospital Avenue, Nedlands, WA 6009, Australia. [15]BC Cancer Agency, 686 West Broadway, Suite 500, Vancouver, BC V5Z 1G1, Canada. [16]Department of Pathology and Laboratory Medicine, Faculty of Medicine, University of British Columbia, Rm. G227–2211 Wesbrook Mall, Vancouver, BC V6T 2B5, Canada. [17]Division of Newborn Medicine, Boston Children's Hospital, 300 Longwood Ave, BCH 3146, Boston, MA 02115, USA. [18]Department of Medical Genetics, University of British Columbia, Vancouver V6T1Z4 BC, Canada. [19]Center for Applied Pediatric Quality Analytics, Boston Children's Hospital, Boston 02115 MA, USA. [20]UBC Centre for Heart and Lung Innovation, Vancouver V6T1Z4 BC, Canada. [21]Department of Medicine, Division of Respiratory Medicine, UBC, Vancouver V6T1Z4 BC, Canada. [22]The Vaccine Centre, Faculty of Infectious and Tropical Diseases, London School of Hygiene and Tropical Medicine, London WC1E 7HT, UK. [23]Broad Institute of MIT & Harvard, Cambridge 02142 MA, USA. [24]Institute for Medical Immunology, Université libre de Bruxelles, Charleroi, Rue Adrienne Bolland 8, 6041 Gosselies, Belgium. [25]Present address: Telethon Kids Institute, 100 Roberts Road, Subiaco 6008, Australia. [26]Present address: Department of Health Science and Technology, Aalborg University, Fredrik Bajers Vej 7 D2, 9220 Aalborg, Denmark. [27]Present address: Department of Pathology, Anatomy and Cell Biology, Thomas Jefferson University, Jefferson Alumni Hall, 1020 Locust Street, Suite 279, Philadelphia, PA 19107, USA. These authors contributed equally: Amy H. Lee, Casey P. Shannon, Nelly Amenyogbe, Tue B. Bennike, Joann Diray-Arce, Olubukola Idoko. These authors jointly supervised this work: Anita H. J. van den Biggelaar, Hanno Steen, Scott J. Tebbutt, Beate Kampmann, Ofer Levy, Tobias R. Kollmann.

## The EPIC Consortium

Diana Vo[5], Ken Kraft[5,6], Kerry McEnaney[5,18], Sofia Vignolo[5] & Arnaud Marchant[24]

