## [Peer Review File · Nature Communications]

Reviewers' comments:

Reviewer #1 (Remarks to the Author):

This manuscript describes a highly valuable dataset on early life development of the immune system, with comprehensive data on transcriptomics, proteomics, metabolomics, cytokines and FACS. It's a remarkable technical achievement, given the difficulty of getting sufficient materials from newborns to perform these assays. As little is known in this important area so far, this dataset deserves a place in a top journal.

The interpretation of the data, however, appears to be half-baked. The authors demonstrated proficient technical ability to perform significance tests and data integration, which were done rather thoroughly. But the biological conclusions are not clear in this manuscript. Most sections are technical validations, but not clear scientific findings a general readership would expect. In the opinion of this reviewer, the supplemental figures are rather more informative than the main figures.

This manuscript would benefit much by anchoring the presentation on a specific and clear immunological finding. The most important question here is the developmental aspect, and the data are clear that temporal trends exist for transcriptomics (Figure 2), cell populations (Suppl Fig 1), cytokines & chemokines (Suppl Fig 2), and metabolites (Suppl Fig 5). Are these trends related? If so, how? The answer is supposed to be given by DIABLO and/or MMRN, but the authors didn't deliver. The title is actually on the money, but the authors need tell us what the "robust developmental trajectory" is.

The data in Suppl Figure 1H, Suppl Figure 2E-F should be leading the main figures. The composition of immune cell populations during the 1st week of life has important implications to the infant response to infection or vaccination. The integration is expected to explain how the cell populations are associated with molecular pathways, and if they are associated with major functional markers. Previous data in the field can supply adult references to compare with, and cell-specific gene signatures to test with. If the authors try to reorganize the data from the most significant components in DIABLO (Figure 4A and 6C) and from MMIN (Figure 5A) on the temporal cell/cytokine axes, the biology will be much more clear and interesting.

A cautionary note on the difference between PCA and PLS. The PC1s in different data types may not reflect the same biology, and PLS is antagonist of that problem. It's not clear if the authors mixed their use, which is not recommended.

A few sloppy errors found in the manuscript:

- The supplemental materials have a mismatched title.
- In [Discussion] especially dependent on innate immunity (ref to Kollmann et al Immunity 2017, already in ref list).
- In [Suppl method] "Liquid nitrogenR"

Reviewer #2 (Remarks to the Author):

The authors present systems biology big data from whole blood obtained from 30 healthy neonates at birth (day of life (DOL) 0), DOL1, DOL3 and DOL7 to delineate ontogenetic changes during the first week of life. They present a standardized processing of low volume blood samples (<1ml) to extract transcriptomic, proteomic, metabolomic, cytokine/chemokine, and single cell immune phenotyping data, apt for the work up of blood samples from newborns. This approach enabled the most comprehensive systems biology study hitherto performed in human newborns. The reader learns that only longitudinal studies and indexing to baseline (here DOL0) can overcome high

inter-individual variances and reveal patterns of developmental changes by performing PCAs from non-curated and normalized immune phenotyping, cytokine and transcriptomic and proteomic data. The authors employed three different strategies of data integration in order to correlate the data from the different platforms and to denominate novel pathways that are likely to be activated or deactivated during the first week of life.

The major claims of this paper are that the expression pattern of whole blood at different molecular levels is characterized by 1) the relief from stress, 2) decreased iron uptake/heme biosynthesis, 3) increased innate immune signaling (Type I interferon, Toll-like receptor and NOTCH signaling), 4) increased complement cascade regulation and 5) metabolic pathway changes like the downregulation of fatty acyl CoA synthesis.

Many aspects of single developmental changes have already been reported before, partly by the authors themselves. The novelty of this study is the comprehensiveness of raised data that allowed to identify many novel candidates of developmental changes, particularly the adding of proteomic and metabolic data. However, for publication in NC there are some major concerns with respect to methods used, form of presentation and work out of the biological relevance of raised data.

Comments

1) In the era of single cell approaches transcriptomic data from whole blood might be inadequate and create difficulties to interpret the data. It remains unclear in what way the identified developmental changes at the transcriptomic, proteomic and cytokine level are due to the known dramatic changes in the composition of blood cells during the first week of life. The authors should deconvolute the OMICs data to distinct cell types for which comprehensive phenotyping has been performed.

2) The strategies of data integration are insufficiently explained, particularly the DIABLO algorithm (What exactly are 'features'? What is 'L1 penalization?'). Why do the different strategies identify different pathways, e.g. why DIABLO does not identify the upregulation of type I interferon signaling?

3) The relations between text and main figures are loose and hamper the readability and understandability of figures and data interpretation (no relation to subpanels). Figure legends insufficiently explain what exactly is shown, especially with respect to the different heatmaps. The manuscript needs thorough revision of the method and result section as well as data presentation in order to reach a broader readership and to allow the reader to draw own conclusions.

4) The major objective to publication in NC is that the findings remain a description of postnatal transcriptomic, proteomic and metabolic phenotypes. Studies addressing to the biological relevance of the identified changes are lacking although the authors conclude that the first week of life is a key for health and disease. Interesting are for example the major inter-individual differences within this cohort of 30 individuals which have not been worked out. Do they correlate with birth-related parameters and metadata on the outcome of health and disease? Are the identified trajectories such robust that they occur independent of individual metadata?

Reviewer #3 (Remarks to the Author):

Review Kollmann and colleagues

This study provides an extensive amount of new 'omic' data related to whole blood in the first week of human life; that contributes profiles of gene expression derived from sequencing technologies, plasma proteomics derived from cytokine-antibody assays and mass spectroscopy (MS) and metabolites from a commercial well validated non-targeted MS/MS platform. This combination of data and longitudinal sampling provides a unique data resource. This is clearly reflected in the manner by which the work has been written up - as essentially a resource paper.

This is challenging work in a challenging population and hence relatively understudied and for this reason is of high significance and interest.

There are few points that need to be addressed.

1. As a resource paper it is essential the data (including primary data) is made available in public repository sites, such as NCBI GEO, in addition to writing to the authors requesting the data.
2. In terms of metabolomic data presentation, it is not clear whether all listed metabolites have been compared directly with an actual standard in their library system, or only some. This should be clearly stated as phrases such as"compounds were identified by comparison to Metabolon library entries of purified standards or recurrent unknown entities" is ambiguous. For example, if only 10% of what is named was identified based on a direct comparison with a standard then the other 90% are only putative and with low confidence that they are correctly identified. A more informative and helpful resource approach would be to state at what level of identification this is in terms of the Metabolomics Standards Initiative levels (<http://cosmos-fp7.eu/msi>). Specifically, what proportion of what metabolites are at each level, and to name those at each level.
3. It is essential that human observational data are appropriately powered to draw conclusions. All approaches used in this study have not been previously reported in the neonatal population, including transcriptional profiling that has only used microarray technology in the past. Therefore, it is important for all methods/data types used that power calculations are provided. (This cannot be performed using the investigated samples) and must be presented with independent sample analysis.
4. Related to the above point on knowing the power of the study sample size is that the authors state that there exists a considerable amount of inter patient variation and indicate that they had to use an indexing approach. Although not clearly written this indexing approach appears to be an appropriate paired analysis of samples (this is not explicitly clear in the manuscript and is inferred on the basis that only two samples are taken from an individual) and thus as expected paired analysis inherently provides greater power. The authors need to be clear about the study design and analysis.
5. In this regard it is important that workflow diagrams showing the n sample number and analysis methodology applied at each stage is provided.
6. The authors mention testing sex as a confounding factor, however, it would be more helpful if a table of all confounding factors tested and outcome of analysis is provided. For this study it should also include when feeding started, first passing of meconium, weight, blood cell proportion changes etc
7. In a number of Figures it is not clear why there are differences in the number of data points shown – if there has been a reduction in the number of samples analysed then this needs to be justified. That is what is the reason for the sample data not being shown. This is OK but needs to be clear and of course impacts on the power for those observational investigations and hence significance of conclusions drawn.
8. For data on the cellular changes it would be helpful to show all cell proportions at the different days including those that do not change (for example T cells that are only assumed not to change from an implicit statement made in the text but the data is not shown)
9. It is notable that the major cell changes occur with myeloid cells (increasing with age). In conjunction with point 6 above – the transcriptomal interferon/inflammasome signature is a hallmark of myeloid cells and therefore it is critical that this increased detected expression signature is not confounded by simply representing an increase in the number of myeloid cells.
10. The authors should clearly state whether D0 is umbilical cord blood or peripheral blood draw from the neonate.

1. Reviewers' comments:

Reviewer #1 (Remarks to the Author):

This manuscript describes a highly valuable dataset on early life development of the immune system, with comprehensive data on transcriptomics, proteomics, metabolomics, cytokines and FACS. It's a remarkable technical achievement, given the difficulty of getting sufficient materials from newborns to perform these assays. As little is known in this important area so far, this dataset deserves a place in a top journal.

The interpretation of the data, however, appears to be half-baked. The authors demonstrated proficient technical ability to perform significance tests and data integration, which were done rather thoroughly. But the biological conclusions are not clear in this manuscript. Most sections are technical validations, but not clear scientific findings a general readership would expect. In the opinion of this reviewer, the supplemental figures are rather more informative than the main figures

RESPONSE: We thank the reviewer for making this important point. Space limitations, and the sheer complexity of the data were the major reasons why the biological conclusions were perhaps less evident than we would have liked. As requested by the reviewer, we have moved the more important observations regarding biological function into the main manuscript and now are providing this more detailed information in several updated and new figures (see e.g. new or updated Figure 2, 3 and 6); we also have emphasized discussion of the associated biological conclusions in the text (see e.g. Abstract Lines 156-159; Results Lines 239-244, 304-308; Discussion Lines 646-666, 677-680).

This manuscript would benefit much by anchoring the presentation on a specific and clear immunological finding. The most important question here is the developmental aspect, and the data are clear that temporal trends exist for transcriptomics (Figure 2), cell populations (Suppl Fig 1), cytokines & chemokines (Suppl Fig 2), and metabolites (Suppl Fig 5). Are these trends related? If so, how? The answer is supposed to be given by DIABLO and/or MMRN, but the authors didn't deliver. The title is actually on the money, but the authors need tell us what the "robust developmental trajectory" is.

RESPONSE: As requested by the reviewer, we have increased the manuscript focus on the biology (including immunobiology) that is driving the developmental trajectory. The relationship of the developmental trends across data types now has also been emphasized in text and figures (see e.g. new Figure 6 and Supplemental Figure 7). In particular, we have provided considerable discussion based on the observation of 3 key pathways determined through meta-integration, for example in lines 646-666: "Our integrative approach identified several key pathways as central to ontogeny. These pathways centered around and interconnected interferon signaling, the complement cascade, and neutrophil activity. Each of these have previously been recognized individually as possible contributors to the increase in newborn susceptibility to infection^{1, 2, 3, 4, 5, 6, 7, 8, 9, 10}, but their age-dependent change had not previously been known to be centrally important to ontogeny over the first week of human life. Importantly, the three key pathways constituting the core of an early life developmental trajectory were readily validated across two independent cohorts of newborns from very different populations, further confirming that the observed early life trajectory is common and predictable thus could serve as a baseline reference. The finding of a stable

trajectory in the first week of life has profound implications: the newborn immune system is still often viewed as 'immature', which implies a stochastic, unregulated state¹¹; however, the existence of the developmental trajectory we discovered as shared by newborns across very different populations in the world strongly argues that early life immune ontogeny is not random but follows a precise and thus possibly purposeful path^{1, 11}. Alternatively, the developmental trajectory we observed over the first week of life could be the result of a limited range of response possibilities in early life, as recently espoused in the hypothesis of stereotypic immune system development¹². Differentiating between these possibilities (a proactive, purposeful path vs. a reactive, restricted response) will be critical for understanding normal development as well as diagnosing, preventing and treating disease in early life¹. This important and novel observation now is also stated clearly in the Abstract.

The data in Suppl Figure 1H, Suppl Figure 2E-F should be leading the main figures. The composition of immune cell populations during the 1st week of life has important implications to the infant response to infection or vaccination. The integration is expected to explain how the cell populations are associated with molecular pathways, and if they are associated with major functional markers. Previous data in the field can supply adult references to compare with, and cell-specific gene signatures to test with. If the authors try to reorganize the data from the most significant components in DIABLO (Figure 4A and 6C) and from MMIN (Figure 5A) on the temporal cell/cytokine axes, the biology will be much more clear and interesting.

RESPONSE: As requested by the reviewer, we have moved the data from supplemental to main figures for cell composition and cytokines (see new Figure 2). To address the impact of changes in cell composition on other OMICS data modalities we first considered that cellular, i.e. RNA (transcriptomic), rather than soluble (plasma) fractions would be most affected by changes in cell composition. We tried multiple approaches to “correcting” for cellular composition differences, but these did not prove to provide any insights. We therefore used Wald statistics in DESeq2 (the program that provides the fold changes in transcript abundance from RNA-Seq) to model if there was any effect of including our observed changes in cell composition. Pearson correlation ($P < 10^{-50}$) of the two models (with and without consideration of cell changes) demonstrated that, overall, transcriptomic signals driving the developmental trajectory were not affected/driven by changes in cell composition. We have emphasized this important insight into the text (Lines 255-265) and the updated Supplementary Figure 2H. Lastly, we added the specific cellular features most significant in *DIABLO* to Figure 6 and Supplementary Figure 7 to provide a focus on the overarching key immunological themes of our findings.

A cautionary note on the difference between PCA and PLS. The PC1s in different data types may not reflect the same biology, and PLS is antagonist of that problem. It's not clear if the authors mixed their use, which is not recommended.

RESPONSE: To address the reviewer's concern we have changed the axis labels when figures represent output of PCA or *DIABLO*, to read “PCX” (principal component 1, 2, ...) or “*DIABLO* Comp. X”, respectively. We agree with the reviewer, that matrix decomposition using e.g. PCA, carried out separately in each dataset is not guaranteed to yield PC1s that reflect the same underlying biological process. Indeed, one of the different integrative methods we applied is meant to address this very issue: While PCA

constructs components maximizing variance in each dataset separately, *DIABLO* (closely related to sparse, generalized canonical correlation analysis) instead constructs components jointly across a set of data matrices by maximizing their covariance to each other, and a vector of outcomes (in our case age [day of life]). In doing so, we identified features in each data matrix that related similarly to outcome and thus were more likely to represent common biological activity related to said outcome (age) that was independently captured across the input data matrices. More specifically, we did not simply 'assume' this to be the case but confirmed it by carrying out the appropriate pathway enrichment analysis.

A few sloppy errors found in the manuscript:

- *The supplemental materials have a mismatched title.*
- *In [Discussion] especially dependent on innate immunity (ref to Kollmann et al Immunity 2017, already in ref list).*
- *In [Suppl method] "Liquid nitrogenR"*

RESPONSE: We appreciate the reviewer's diligent reading of the manuscript and have now corrected these typos.

Reviewer #2 (Remarks to the Author):

The authors present systems biology big data from whole blood obtained from 30 healthy neonates at birth (day of life (DOL) 0), DOL1, DOL3 and DOL7 to delineate ontogenetic changes during the first week of life. They present a standardized processing of low volume blood samples (<1ml) to extract transcriptomic, proteomic, metabolomic, cytokine/chemokine, and single cell immune phenotyping data, apt for the work up of blood samples from newborns. This approach enabled the most comprehensive systems biology study hitherto performed in human newborns. The reader learns that only longitudinal studies and indexing to baseline (here DOL0) can overcome high inter-individual variances and reveal patterns of developmental changes by performing PCAs from non-curated and normalized immune phenotyping, cytokine and transcriptomic and proteomic data. The authors employed three different strategies of data integration in order to correlate the data from the different platforms and to denominate novel pathways that are likely to be activated or deactivated during the first week of life. The major claims of this paper are that the expression pattern of whole blood at different molecular levels is characterized by 1) the relief from stress, 2) decreased iron uptake/heme biosynthesis, 3) increased innate immune signaling (Type I interferon, Toll-like receptor and NOTCH signaling), 4) increased complement cascade regulation and 5) metabolic pathway changes like the downregulation of fatty acyl CoA synthesis. Many aspects of single developmental changes have already been reported before, partly by the authors themselves. The novelty of this study is the comprehensiveness of raised data that allowed to identify many novel candidates of developmental changes, particularly the adding of proteomic and metabolic data. However, for publication in NC there are some major concerns with respect to methods used, form of presentation and work out of the biological relevance of raised data.

Comments

1) In the era of single cell approaches transcriptomic data from whole blood might be inadequate and create difficulties to interpret the data. It remains unclear in what way the identified developmental changes at the transcriptomic, proteomic and cytokine level

are due to the known dramatic changes in the composition of blood cells during the first week of life. The authors should deconvolute the OMICS data to distinct cell types for which comprehensive phenotyping has been performed.

RESPONSE: We agree with the reviewer but would like to add that every method has its advantages and disadvantages, including the single cell OMICS approaches (see for example the entire issue of *Nature Reviews Genetics*, January 2018). More specifically, whole blood transcriptomic approaches are still very informative (e.g. ¹³), especially as they allow the necessary processing on site in field studies of low resource settings (i.e. no special equipment required), which is precisely what our study emphasizes. Furthermore, and as outlined in response to Reviewer 1 above, we have assessed if cell composition impacts our OMICS (especially transcriptomic) data in several ways and shown that it did not confound the final result. For example, as mentioned above we used Wald statistics in DESeq2 to demonstrate by Pearson correlation ($P < 10^{-50}$) that there were no major effects of our observed changes in transcriptomics that could be explained by changes in cell composition (Supplementary Figure 2H).

2) *The strategies of data integration are insufficiently explained, particularly the DIABLO algorithm (What exactly are 'features'? What is 'L1 penalization'?). Why do the different strategies identify different pathways, e.g. why DIABLO does not identify the upregulation of type I interferon signaling?*

RESPONSE: We thank the reviewer for this comment, as our response will ensure we are as clear as is possible. We have now removed the complicated and ambiguous word 'features' and replaced this with 'markers' in the *DIABLO* analysis and 'nodes' in the OMICS based analyses. We have also clarified in the text that these terms refer to cells, transcript, proteins, metabolites (i.e. the data types being integrated). We have also removed the mention of L1 penalization which describes a technical process, instead now simply state that the method is capable of carrying out marker selection to identify a minimal subset of cells, transcripts, proteins, metabolites that relate to age (DOL). Additional detail is now also provided in the "Methods (Online)" section, as well as in the cited *DIABLO* publication and on the mixOmics R package website (which is also cited).

As to why different strategies identified not only the same but also different pathways, we would like to point out that one of our key findings was that there was indeed substantial overlap between analytical approaches, which we have further emphasized in the section on Meta-Integration to clarify that point. For example, the interferon pathway was identified by *DIABLO* as well as our two other analytical approaches. Specifically, identification of different markers and pathways by different analytical methods was what would be expected given the different analytical algorithms, with each following a different underlying strategy. The fact that any overlap between these vastly different approaches existed at all – and indeed in our study this overlap was substantial- and moreover focused on a coherent biological concept (e.g. interferon pathway, neutrophil signals, and complement) constitutes one of our key findings.

3) *The relations between text and main figures are loose and hamper the readability and understandability of figures and data interpretation (no relation to subpanels). Figure legends insufficiently explain what exactly is shown, especially with respect to*

the different heatmaps. The manuscript needs thorough revision of the method and result section as well as data presentation in order to reach a broader readership and to allow the reader to draw own conclusions.

RESPONSE: In response to this helpful feedback, we have changed several of the Figures (as mentioned above) to provide additional support for key concepts mentioned in the text, and in particular reworked the manuscript as requested, emphasizing the links between Figures and the text; we also enhanced the information provided in each of the Figure legends.

4) The major objective to publication in NC is that the findings remain a description of postnatal transcriptomic, proteomic and metabolic phenotypes. Studies addressing to the biological relevance of the identified changes are lacking although the authors conclude that the first week of life is a key for health and disease. Interesting are for example the major inter-individual differences within this cohort of 30 individuals which have not been worked out. Do they correlate with birth-related parameters and metadata on the outcome of health and disease? Are the identified trajectories such robust that they occur independent of individual metadata?

RESPONSE: We thank the reviewer for bringing up this important topic. First it is worth mentioning that the multi-omics and data integration approaches provided here confer immediate cross validation of the biological findings. Thus, we correlate significant changes in gene expression, plasma proteome and metabolome, cell types and cytokines to confirm both the conserved and novel pathways identified here (see Figures 2-6). Secondly, we did not design this study to correlate birth-related parameters and metadata with outcome of health and disease for a given individual, but to extract the key factors that drive newborn development in most newborns. However, what we do identify are several consistently influential pathways (eg, interferon, complement, neutrophil function) that are known to be relevant to health and disease for newborns in general. Most importantly, to address the issue of biological relevance, we have added a validation cohort of 30 infants from a different genetic and environmental background to test if the model predicted to drive this striking developmental trajectory based on data from Gambian newborns, would also hold true. Indeed, the core elements of the trajectory were confirmed across both the Gambian and Papua New Guinean cohorts, strongly emphasizing the general biological relevance of our findings (please see text and new Figure 6, as well as Supplemental Figures 8 and 9). We are intrigued by our findings of a robust core developmental trajectory in newborns akin to the stable adult baseline. These findings lay the basis, per the Reviewer's helpful feedback, to conduct future studies, beyond the scope of this current manuscript, comprised of large cohorts followed up over years, aimed at delineating the impact of unique personal life events (birth mode, nutrition, infection, vaccination etc.) on this trajectory.

Reviewer #3 (Remarks to the Author):

This study provides an extensive amount of new 'omic' data related to whole blood in the first week of human life; that contributes profiles of gene expression derived from sequencing technologies, plasma proteomics derived from cytokine-antibody assays

and mass spectroscopy (MS) and metabolites from a commercial well validated non-targeted MS/MS platform. This combination of data and longitudinal sampling provides a unique data resource. This is clearly reflected in the manner by which the work has been written up - as essentially a resource paper. This is challenging work in a challenging population and hence relatively understudied and for this reason is of high significance and interest.

RESPONSE: We thank the reviewer for his insightful comments. We do indeed see this as a resource paper but in response to the other reviewer's comments we have also tried to improve the biological narrative.

There are few points that need to be addressed.

1. As a resource paper it is essential the data (including primary data) is made available in public repository sites, such as NCBI GEO, in addition to writing to the authors requesting the data.

RESPONSE: We now have provided access to the public data bases for all of our data, see Online Methods Line 1022-1026. Moreover, as our study is in part support by NIH (NIAID)'s Human Immunology Project Consortium, we will be depositing the data from this manuscript into *ImmPort*, an NIH-supported public website for data sharing.

2. In terms of metabolomic data presentation, it is not clear whether all listed metabolites have been compared directly with an actual standard in their library system, or only some. This should be clearly stated as phrases such as"compounds were identified by comparison to Metabolon library entries of purified standards or recurrent unknown entities" is ambiguous. For example, if only 10% of what is named was identified based on a direct comparison with a standard then the other 90% are only putative and with low confidence that they are correctly identified. A more informative and helpful resource approach would be to state at what level of identification this is in terms of the Metabolomics Standards Initiative levels (<http://cosmos-fp7.eu/msi>). Specifically, what proportion of what metabolites are at each level, and to name those at each level.

RESPONSE: We have extensively edited the metabolomics section in Methods according to the reviewer's suggestion. We have, according to the reporting standards stated by the Chemical Analysis Working Group (CAWG) of the Metabolomics Standards Initiative (MSI) now categorized all Level 1 metabolites with retention time/index with mass spectrum, accurate mass and tandem MS.

3. It is essential that human observational data are appropriately powered to draw conclusions. All approaches used in this study have not been previously reported in the neonatal population, including transcriptional profiling that has only used microarray technology in the past. Therefore, it is important for all methods/data types used that power calculations are provided. (This cannot be performed using the investigated samples) and must be presented with independent sample analysis.

RESPONSE: We agree with the reviewer, that the optimal approach to a priori power calculations in systems biology is difficult in the absence of knowledge about effect size, and that the best approach is validation of findings in a subsequent, appropriately powered independent cohort. Per the Reviewer's request, in the updated manuscript we now include validation of the key findings made in newborns from The Gambia in an

appropriately powered, independent cohort of newborns from Papua New Guinea. Given we had an estimated effect size from our training data set in The Gambia, we estimated the sample size needed for our PNG validation cohort based on The Gambia cohort data (i.e. effect size of age). Specifically, using the approach described by Peng et al.¹⁴ the mean fold-change and standard deviation were estimated in the Gambia transcriptomic data (using the data provided in our Supplemental Tables), and, controlling the FDR at 5%, under various assumptions for the proportion of non-DE features (0.8, 0.9), once again based on what we observed in our Gambian cohort (~10% of transcripts DE at DOL3 and ~15% of transcripts DE at DOL7) we estimated that with $n = 10$, we would have >80% power even assuming a very low proportion of DE features (10%). This is now included in Online methods. See the graph below for the calculations.

4. Related to the above point on knowing the power of the study sample size is that the authors state that there exists a considerable amount of inter patient variation and indicate that they had to use an indexing approach. Although not clearly written this indexing approach appears to be an appropriate paired analysis of samples (this is not explicitly clear in the manuscript and is inferred on the basis that only two samples are taken from an individual) and thus as expected paired analysis inherently provides greater power. The authors need to be clear about the study design and analysis.

RESPONSE: We thank the reviewer for highlighting this important aspect of our data analysis approach. The reviewer is correct that paired analysis of samples was carried out for all univariate test results. For multivariate approaches (including PCA and *DIABLO*), the sample pairing was addressed explicitly by decomposing the variance in the dataset into the between- and within- subject variation as previously described¹⁵ and subsequently focused on operating on the within-subject variation matrix. To clarify this important point, we have moved a paragraph describing this into the Results section Lines 201-213, clarifying this as: "This required indexing either by employing paired statistical tests for univariate analyses or calculating fold changes relative to the DOL0 sample for multivariate analyses...". The more complete description for this is now included in a section in online methods.

5. In this regard it is important that workflow diagrams showing the *n* sample number and analysis methodology applied at each stage is provided.

RESPONSE: We now provide this workflow information in Figure 1 for the Gambia cohort and in Supplemental Figure 8 for the PNG cohort.

6. The authors mention testing sex as a confounding factor, however, it would be more helpful if a table of all confounding factors tested and outcome of analysis is provided. For this study it should also include when feeding started, first passing of meconium, weight, blood cell proportion changes etc

RESPONSE: We fully agree with the reviewer of the importance of these aspects that should be addressed in future studies that will benefit from our paper as a resource. For now we note that the important issues regarding the impact specific of parameters beyond sex such as genetics, when feeding started, when meconium was passed, weight, etc., was in fact mitigated by indexing and we thus avoided having to address these individual issues in the current study.

7. In a number of Figures it is not clear why there are differences in the number of data points shown – if there has been a reduction in the number of samples analysed then this needs to be justified. That is what is the reason for the sample data not being shown. This is OK but needs to be clear and of course impacts on the power for those observational investigations and hence significance of conclusions drawn.

RESPONSE: We have now provided this information in Figure 1 for the Gambia cohort and in Supplemental 8 for the PNG cohort and provide the accompanying detailed information in the text (Supplementary Text, Lines 18-26).

8. For data on the cellular changes it would be helpful to show all cell proportions at the different days including those that do not change (for example T cells that are only assumed not to change from an implicit statement made in the text but the data is not shown) .

RESPONSE: We agree with the reviewer and now provide this data in Figure 2 as well as in Supplemental Figure 1.

9. It is notable that the major cell changes occur with myeloid cells (increasing with age). In conjunction with point 6 above – the transcriptomal interferon/inflammasome signature is a hallmark of myeloid cells and therefore it is critical that this increased detected expression signature is not confounded by simply representing an increase in the number of myeloid cells.

RESPONSE: We have addressed the impact of changes in cell composition on OMICs data output in the responses to the other reviewer's above. Specifically, we used Wald statistics in DESeq2 (the program that provides the fold changes in transcript abundance from RNA-Seq) to model if there was any effect of our observed changes in cell composition. Pearson correlation ($P < 10^{-50}$) of the two models (with and without consideration of cell changes) demonstrated that, overall, transcriptomic signals driving the developmental trajectory were not affected/driven by changes in cell composition (Supplementary Figure 2H).

10. The authors should clearly state whether D0 is umbilical cord blood or peripheral blood draw from the neonate.

RESPONSE: We thank the reviewer for this point as this is indeed an important distinction. All of our samples, including DOL0 were peripheral blood. We now have made this clear in the text (Results and Methods).

Overall the extensive changes and new data have further enhanced our manuscript which we trust is now acceptable at *Nature Communications*.

Sincerely,
Tobias R. Kollmann & Ofer Levy

REFERENCES

1. Kollmann, T.R., Kampmann, B., Mazmanian, S.K., Marchant, A. & Levy, O. Protecting the Newborn and Young Infant from Infectious Diseases: Lessons from Immune Ontogeny. *Immunity* **46**, 350-363 (2017).
2. Radtke, F., Wilson, A., Mancini, S.J. & MacDonald, H.R. Notch regulation of lymphocyte development and function. *Nat Immunol* **5**, 247-253 (2004).
3. Abdelhaleem, M. RNA helicases: regulators of differentiation. *Clin Biochem* **38**, 499-503 (2005).
4. Loo, Y.M. & Gale, M., Jr. Immune signaling by RIG-I-like receptors. *Immunity* **34**, 680-692 (2011).
5. Schmidt, C.Q., Lambris, J.D. & Ricklin, D. Protection of host cells by complement regulators. *Immunol Rev* **274**, 152-171 (2016).
6. Aksoy, E. *et al.* Interferon regulatory factor 3-dependent responses to lipopolysaccharide are selectively blunted in cord blood cells. *Blood* **109**, 2887-2893 (2007).
7. Danis, B. *et al.* Interferon regulatory factor 7-mediated responses are defective in cord blood plasmacytoid dendritic cells. *Eur. J. Immunol.* **38**, 507-517 (2008).
8. Kollmann, T.R. *et al.* Neonatal Innate TLR-Mediated Responses Are Distinct from Those of Adults. *The Journal of Immunology* **183**, 7150-7160 (2009).
9. Corbett, N.P. *et al.* Ontogeny of Toll-like receptor mediated cytokine responses of human blood mononuclear cells. *PLoS One* **5**, e15041 (2010).
10. Lawrence, S.M., Corriden, R. & Nizet, V. Age-Appropriate Functions and Dysfunctions of the Neonatal Neutrophil. *Frontiers in pediatrics* **5**, 23 (2017).
11. Harbeson, D., Ben-Othman, R., Amenyogbe, N. & Kollmann, T.R. Outgrowing the Immaturity Myth: The Cost of Defending From Neonatal Infectious Disease. *Frontiers in immunology* **9**, 1077 (2018).
12. Olin, A. *et al.* Stereotypic Immune System Development in Newborn Children. *Cell* **174**, 1277-1292.e1214 (2018).
13. Urrutia, A. *et al.* Standardized Whole-Blood Transcriptional Profiling Enables the Deconvolution of Complex Induced Immune Responses. *Cell reports* **16**, 2777-2791 (2016).

14. Liu, P. & Hwang, J.T. Quick calculation for sample size while controlling false discovery rate with application to microarray analysis. *Bioinformatics* **23**, 739-746 (2007).
15. Westerhuis, J.A., van Velzen, E.J., Hoefsloot, H.C. & Smilde, A.K. Multivariate paired data analysis: multilevel PLSDA versus OPLSDA. *Metabolomics : Official journal of the Metabolomic Society* **6**, 119-128 (2010).

Reviewers' comments:

Reviewer #1 (Remarks to the Author):

I think the authors have adequately addressed all concerns from the three reviewers.

The question raised by reviewer #2 on bulk transcriptomics vs single-cell sequencing is not a settled debate. I think the authors fully demonstrated the value of the bulk transcriptomics data. As such data will be common on clinical samples in the near future, the approach and conclusion from this paper will remain highly relevant.

On the question of cell types compounding the molecular omics: I personally do not think the deconvolution methods are necessary or good enough, as long as the interpretation is not simplified or overlooked. The omics data deliver more information than the composition of cell populations.

The power calculation requested by reviewer #3 is an open question. Because pathway organization and concordance between multiple -omics introduce additional information into the data analysis, the traditional feature-level power calculation is helpful but not entirely authoritative, rather prone for false negatives. I agree with the authors that this work has presented a fair treatment of systems analysis.

The manuscript now has a clear biological narrative that adds to our knowledge landscape. Importantly, the addition of data from the PNG cohort truly solidifies the findings.

Reviewer #3 (Remarks to the Author):

The revised manuscript is much improved and certainly addresses a number of my concerns. However there remains a few issues and in some cases an incomplete response to a number of those concerns that should be addressed.

point 1 -- addressed

point 2 response addresses the use of the metabolomics standard initiative. However, it would be helpful in this context to include a supplementary table of all the standards used (3,300) as this information is not readily available.

point 3 response addressed the power calculations for the transcriptomic data but does not include power calculations for the proteomic or metabolite data. These power calculations are also essential as the study attempts to draw conclusions from those data as well.

point 4 response is satisfactory but for systems analysis a chart of the precise work flows should be provided in the supplementary section as well. See comments to point 5 below.

point 5 -- the response now includes a new figure (suppl. Fig 8) that provides information on how many samples were used/analysed for the different platforms. This is NOT a work flow chart and it is critical this should be provided. Work flows should include the processing/parameterisation, statistical and bioinformatics analyses steps as applied to the data -- the picture in supply.Fig8 provides minimal information although quite pretty with colours -- black and white flow diagrams would be sufficient.

point 6 the response attempts to argue that potential confounders are mitigated. This is incorrect as they cannot be mitigated if not measured. If this data is available then an analyses should be

conducted or if the data is not available then the authors should state that this study does not account for such potential confounders and which may underly some of the variation.

point 7 The response is not completely satisfactory -- however detailed work flow diagrams as described above would clarify this concern.

point 8 - satisfactory response is provided

point 9 The authors have now applied a Wald test to address this point. This is a partly satisfactory response. This is a parametric test and assumes normality of the data. However, from the volcano plots the log transformed data for DOL0/3 (which would be anticipated to be normal after transforming) appears not to be normal. For this test to be valid the authors need to test for normality. Alternatively or in addition, a non-parametric test can be applied that is invariant to re-parametrisation or transformation of the data. Further the authors state they have used adjusted data for accounting for the cell changes but fail to provide how exactly they made this adjustment. It is quite critical to be stringent on this point as a major conclusion of the study is derived from this test.

point10 - I thank the authors for this clarification for not using cord blood as DOL0 -- it would be of interest to know the time window (how many hours/minutes after birth) this sample was taken. Also not consistent with this response is the labelling of data in suppl. Fig2A that labels neonatal blood as "cord". The authors should state that in the case of those data/experiments cord blood was used instead of DOL0.

The original reviewer's comments are in italics with our responses in normal type face:

Reviewers' comments:

Reviewer #1 (Remarks to the Author):

I think the authors have adequately addressed all concerns from the three reviewers.

The question raised by reviewer #2 on bulk transcriptomics vs single-cell sequencing is not a settled debate. I think the authors fully demonstrated the value of the bulk transcriptomics data. As such data will be common on clinical samples in the near future, the approach and conclusion from this paper will remain highly relevant.

On the question of cell types compounding the molecular omics: I personally do not think the deconvolution methods are necessary or good enough, as long as the interpretation is not simplified or overlooked. The omics data deliver more information than the composition of cell populations.

The power calculation requested by reviewer #3 is an open question. Because pathway organization and concordance between multiple -omics introduce additional information into the data analysis, the traditional feature-level power calculation is helpful but not entirely authoritative, rather prone for false negatives. I agree with the authors that this work has presented a fair treatment of systems analysis.

The manuscript now has a clear biological narrative that adds to our knowledge landscape. Importantly, the addition of data from the PNG cohort truly solidifies the findings.

Response: We thank the reviewer for these helpful assessments and for the views expressed above, especially those regarding cell types and power calculations, which entirely reflect and endorse our approach.

Reviewer #3 (Remarks to the Author):

The revised manuscript is much improved and certainly addresses a number of my concerns. However there remains a few issues and in some cases an incomplete response to a number of those concerns that should be addressed.

Response: We are delighted to have addressed the vast majority of the reviewer's concerns in our previous revision and appreciate the reviewer's remaining concerns.

point 1 – addressed

point 2 response addresses the use of the metabolomics standard initiative. However, it would be helpful in this context to include a supplementary table of all the standards used (3,300) as this information is not readily available.

Response: We appreciate the reviewer's suggestion and as the reviewer has noted, we included the retention time index, mass of biochemical, HMDB and KEGG ID for all metabolites identified in our experimental data set as proposed by the Metabolomics Standard Initiative (<http://www.metabolomics-msi.org/>), i.e. all standard reporting requirements for discovery metabolomics studies were fulfilled. Of note: In this study, we partnered with Metabolon, one of the leading companies in the field of metabolomics (PMID: 28263315; PMID: 30217994). As has been the case in other high impact published studies in which this company has participated (PMID: 24816252; PMID: 2936236), Metabolon does not share their entire library of standards as they view this information as proprietary. We are therefore unable to list all the standards used.

point 3 response addressed the power calculations for the transcriptomic data but does not include power calculations for the proteomic or metabolite data. These power calculations are also essential as the study attempts to draw conclusions from those data as well.

Response: We are mindful of Reviewer 1's comments regarding power calculations. To address the comment made by Reviewer 3, we have added power calculations for the proteomic and metabolomic datasets. As for the transcriptomic data set, the power calculations were performed using 'ssize.fdr', a conservative 2 fold-change cut-off and standard deviations estimated from our Gambia data for each OMIC data type. As our analyses employed a paired comparison, the standard deviations are derived from the differences of later DOL compared to DOL0. Based on this analysis (see attached Figures below), and the number of samples available in our validation cohort (PNG, vertical line), we observed that both transcriptomics and metabolomics were well powered for the PNG data set (80% power to detect age-dependent differences with $n = 6$), and only the proteomic data was not. However, this is unlikely to have had an impact on our important integration analysis as most of the features used as inputs to *NetworkAnalyst* were strictly derived from the two well-powered OMICs data: transcriptomic or metabolomics. Furthermore, since lack of power increases the risk of Type II errors (false negatives), we would not expect the pathways identified from the proteomics data to be false positives and indeed we note that in our integration studies we obtained good coherence between all three omics in terms of conclusions. To emphasize this further, we now have included this information in the Supplemental Text (pg 3, lines 110 – 113) as follows: "While each of the OMIC data types that we examined revealed statistically significant changes over the first week of life, we used as seed nodes only transcriptomic or metabolomic data to construct PPI-networks in *NetworkAnalyst* as they represented the two highest-powered OMICs data."

point 4 response is satisfactory but for systems analysis a chart of the precise work flows should be provided in the supplementary section as well. See comments to point 5 below.

point 5 -- the response now includes a new figure (suppl. Fig 8) that provides information on how many samples were used/analysed for the different platforms. This is NOT a work flow chart and it is critical this should be provided. Work flows should include the processing/parameterisation, statistical and bioinformatics analyses steps as applied to the data -- the picture in supply.Fig8 provides minimal information although quite pretty with colours -- black and white flow diagrams would be sufficient.

point 7 The response is not completely satisfactory -- however detailed work flow diagrams as described above would clarify this concern.

Response: As suggested by the reviewer and in response to Points #4, #5 and #7, we have a) updated Figure 1 by adding a new Table to list the samples/analysis type more clearly) and b) included a **new Supplementary Figure 1** (as indicated in the Main Text Methods section, lines 510-512), which details the workflow diagram and includes the sample processing procedure as well as the statistical and bioinformatic tools used. Note that with the addition of this new workflow figure as the new Supplementary Figure 1, the numbering for all other Supplementary Figures changed (by +1).

point 6 the response attempts to argue that potential confounders are mitigated. This is incorrect as they cannot be mitigated if not measured. If this data is available then an analyses should be conducted or if the data is not available then the authors should state that this study does not account for such potential confounders and which may underly some of the variation.

Response: The reviewer correctly notes that any analysis should consider potential confounding influences. In our setting, these include sex, recruitment season, time of blood draw, gestational age, birthweight, age at initiation of feeding, first passing of meconium, blood cell proportion changes etc. that could impact the observed effects. We fully recognize that these are important factors to consider and have taken several approaches to minimize the potential influence on our analysis and conclusions. We wish to emphasize two points related to this important issue:

- 1) Because our analyses used within-subject paired tests exclusively, we maintained implicit control of any confounding for covariates. Specifically, comparing each subject to themselves (indexing) minimized variation between subjects since both DOL0 (the baseline) and DOL1/3/7 (the experimental samples) came from the same participant in which all of the factors/confounders listed above would be identical. Thus, we controlled for confounding via study design. This was particularly important and likely explains why we were able to observe similar statistically significant ontogenic patterns over the first week of life across two independent and completely different (i.e. including likely differences in genetics) cohorts from the Gambia and PNG.
- 2) Furthermore, all infants were recruited over a very short time period (1 month) thus seasonality was not an issue. Equally important, all infants were full-term, of normal birth weight, and exclusively breast-fed reducing these as possible confounders. Lastly, peripheral blood was drawn during very tight windows on day of life 0 (DOL0) and processed in the lab within 4 hours reducing the likelihood processing-related confounders impacting our analysis.

To address the reviewers concern, we added a statement to the Main Text listing potential confounders (lines 552-557): “While the limited number of samples available did not allow us to consider additional potential confounders, such as when feeding started, first passing of meconium or birth weight, our approach of indexing data longitudinally for each newborn allowed us to look for consistent differences between each DOL.” Nevertheless, we fully agree with the Reviewer and accept that by definition we cannot know the effect of unmeasured confounders and have thus added the statement: “While other unforeseen confounders may exist, they did not obscure the strong developmental patterns in the data observed during the first week of life across two very different cohorts.”

point 8 - satisfactory response is provided

point 9 The authors have now applied a Wald test to address this point. This is a partly satisfactory response. This is a parametric test and assumes normality of the data. However, from the volcano plots the log transformed data for DOL0/3 (which would be anticipated to be normal after transforming) appears not to be normal. For this test to be valid the authors need to

test for normality. Alternatively or in addition, a non-parametric test can be applied that is invariant to re-parametrisation or transformation of the data. Further the authors state they have used adjusted data for accounting for the cell changes but fail to provide how exactly they made this adjustment. It is quite critical to be stringent on this point as a major conclusion of the study is derived from this test.

Response: We thank the reviewer for this comment. RNA-Seq raw counts data are known to follow a negative binomial distribution (PMID:19910308 and 25516281). The authors of the *DESeq2* (cited by 6586 papers as of Oct 16, 2018) demonstrated that the Wald test is appropriate for hypothesis testing of RNA-Seq data, stating: “the shrunken estimate of LFC is divided by its standard error, resulting in a z-statistic, which is compared to a standard normal distribution.” (PMID: 25516281). Furthermore, we do not feel we can estimate the RNA-Seq count distribution with such a small sample size. We fully agree with the reviewer that it is important to be specific and stringent on how we accounted for cell changes. Therefore, we elaborated on this Main Text Methods in lines 630-639. “To test whether changes in cell composition could account for the observed changes in gene expression, we used *DESeq2* (with default parameters) to fit two models, one including subject and DOL and the other model including the additional covariate of cell composition. To address the collinearity of this compositional data, we used Principal Component Analysis, summarizing the cell proportions (flow cytometry) to 5 principle components (PCs, accounting for 95% of the variance observed). We compared the estimated effect sizes for the DOL term between these two models (Pearson correlation) across all genes and found them to be highly correlated ($p < 10^{-50}$), indicating that the observed changes in gene expression could not be (fully) explained by changes in the underlying cell composition.”

point10 - I thank the authors for this clarification for not using cord blood as DOL0 -- it would be of interest to know the time window (how many hours/minutes after birth) this sample was taken. Also not consistent with this response is the labelling of data in suppl. Fig2A that labels neonatal blood as "cord". The authors should state that in the case of those data/experiments cord blood was used instead of DOL0.

RESPONSE: Thank you to the careful reading of this reviewer. We have explicitly changed the figure legend for Fig2A to state “cord” as well as in the Supplementary Text, page 2, lines 41. Regarding the time window of the blood draws, we refer to our response to Point 6 above as this question directly relates to possible confounders impacting our study results.

REVIEWERS' COMMENTS:

Reviewer #3 (Remarks to the Author):

The authors have provided considered responses.

Responses to point 4-10 are all acceptable.

Response to point 2 is not acceptable. In open source research - data must be reproducible and re-usable. For the metabolism data this requires declaring standards used. With out this information the data is not fully re-usable nor evaluative. I fail to see from a business point of view the proprietary nature of the refusal to show what standards were used in the study. If this company refuses to declare this information then in my opinion they should not undertake work for academic-public funded projects.

For the response to Point 3 there is one small comment - Although not formally correct the retrospective power calculation are informative and show that while the transcriptomic and metabolomic studies maybe OK -- the proteomic is not. There should be a statement of these retrospective power studies (provided in the supplementary) and would help address any future criticism on the study - as well as help temper the overall robustness of the data. Outside this comment, the response on this matter is acceptable,

REVIEWER REQUESTS

Reviewer #3 (Remarks to the Author):

**The authors have provided considered responses. Responses to point 4-10 are all acceptable.*

RESPONSE: We greatly appreciate the reviewer's efforts and are happy that nearly all of our responses have satisfied the requests made.

**Response to point 2 is not acceptable. In open source research - data must be reproducible and re-usable. For the metabolism data this requires declaring standards used. Without this information the data is not fully re-usable nor evaluative. I fail to see from a business point of view the proprietary nature of the refusal to show what standards were used in the study. If this company refuses to declare this information then in my opinion they should not undertake work for academic-public funded projects.*

RESPONSE: As we had already outlined in our response to this point in the previous point-by-point response, we do not have access to this library from Metabolon. But as you know, we have complied with all currently existing data reporting standards in the field of discovery metabolomics research namely the Metabolomics Standards Initiative. Specifically, the metabolites presented in our paper were identified based on relative quantification for each sample processed and we report these metabolites in our paper with retention time index, mass of biochemical, HMDB and KEGG ID. These metabolites were identified by comparison to a library based on authenticated standards that contained the retention time/index (RI), mass to charge ratio (m/z), and chromatographic data (including MS/MS spectral data) on all molecules present in the Metabolon library. Furthermore, Nature Research's own journals such as *Nature* (PMID: 25043030), *Nature Medicine* (PMID: 24710377), *Nature Genetics* ((PMID: 28263315; and PMID: 30217994), *Nature Biotechnology* (PMID: 25850038) as well as *Nature Communications* itself (PMID:29362361; PMID: 26289811; PMID: 25809635) have published research data obtained in collaboration with Metabolon providing only the information we provide here. We thus hope you agree with our current approach to reporting our data.

**For the response to Point 3 there is one small comment - Although not formally correct the retrospective power calculation are informative and show that while the transcriptomic and metabolomic studies maybe OK -- the proteomic is not. There should be a statement of these retrospective power studies (provided in the supplementary) and would help address any future criticism on the study - as well as help temper the overall robustness of the data. Outside this comment, the response on this matter is acceptable.*

RESPONSE: We already had included this information in the Supplemental Text during our previous response. To now comply with this reviewer's additional request, we added the following sentences (pg 3, lines 114 – 125): "Specifically both transcriptomics and metabolomics were well powered for the PNG data set (80% power to detect age-dependent differences with $n = 6$); only the proteomic data was not. However, this was unlikely to have any impact on our integration analysis as the features used as inputs to *NetworkAnalyst* were strictly derived from the two well-powered OMICs data (i.e. transcriptomic and metabolomics). Furthermore, since lack of power increases the risk of Type II errors (false negatives), we would not expect the pathways identified from the proteomics data to be false positives; indeed, we note that in our integration studies we obtained good coherence between all three omics in terms of conclusions".